# Neural Tangent Kernel Analysis and Filtering for Robust Fourier Features Embedding

## Abstract

Implicit Neural Representations (INRs) employ neural networks to represent continuous functions by mapping coordinates to the corresponding values of the target function, with applications e.g., inverse graphics. However, INRs face a challenge known as spectral bias when dealing with scenes containing varying frequencies. To overcome spectral bias, the most common approach is the Fourier features-based methods such as positional encoding. However, Fourier features-based methods will introduce noise to output, which degrades their performances when applied to downstream tasks. In response, this paper addresses this problem by first investigating the underlying causes through the lens of the Neural Tangent Kernel. Through theoretical analysis, we propose that using Fourier features embedding can be interpreted as fitting Fourier series expansion of the target function, from which we find that it is the insufficiency in the finitely sampled frequencies that causes the generation of noisy outputs. Leveraging these insights, we introduce bias-free MLPs as an adaptive linear filter to locally suppress unnecessary frequencies while amplifying essential ones by adjusting the coefficients at the coordinate level. Additionally, we propose a line-search-based algorithm to adjust the filter's learning rate dynamically, achieving Pareto efficiency between the adaptive linear filter module and the INRs. Extensive experiments demonstrate that our proposed method consistently improves the performance of INRs on typical tasks, including image regression, 3D shape regression, and inverse graphics. The full code will be publicly available.

## 1 Introduction

Implicit Neural Representations (INRs), which fit the target function using only input coordinates, have recently gained significant attention. By leveraging the powerful fitting capability of Multilayer Perceptrons (MLPs), INRs can implicitly represent the target function without requiring their analytical expressions. The versatility of MLPs allows INRs to be applied in various fields, including inverse graphics (Mildenhall et al., 2021; Barron et al., 2023; Martin-Brualla et al., 2021), image super-resolution (Chen et al., 2021b; Yuan et al., 2022; Gao et al., 2023), image generation (Skorokhodov et al., 2021), and more (Chen et al., 2021a; Strümpler et al., 2022; Shue et al., 2023).

However, MLPs face a significant challenge known as the spectral bias, where low-frequency signals are typically favored during training (Rahaman et al., 2019). A common solution is to project coordinates into the frequency domain using Fourier features, such as Random Fourier Features and Positional Encoding, which can be understood as manually set high-frequency correspondence prior to accelerating the learning of high-frequency targets. (Tancik et al., 2020). This projection is widely applied in the INRs for novel view synthesis (Mildenhall et al., 2021; Barron et al., 2021), dynamic scene reconstruction (Pumarola et al., 2021), object tracking (Wang et al., 2023), and medical imaging (Corona-Figueroa et al., 2022). Although many INRs' downstream application scenarios use this encoding type, it has certain limitations when applied to specific tasks. It depends heavily on two key hyperparameters: the sampling variance/scale (available sampling range of frequencies) and the number of samples. Varying the sampling variance/scale may lead to degradation results, as shown in Figure 1. Even with a proper choice of sampling variance/scale, the output remains unsatisfactory, as shown in Figure 2: Noisy low-frequency regions and degraded high-frequency regions persist with well chosen sampling variance/scale with the grid-searched variance/scale, which may potentially affect the performance of the downstream applications resulting in noisy or coarse

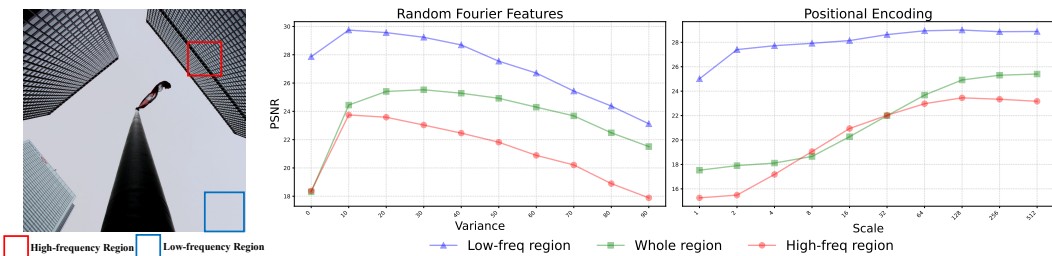

Figure 1: We test the performance of MLPs with Random Fourier Features (RFF) and MLPs with Positional Encoding (PE) on a 1024-resolution image to better distinguish between high- and low-frequency regions, as demonstrated on the left-hand side of this figure. We find that the performance of MLPs+RFF degrades rapidly with increasing variance, while MLPs+PE, although it doesn't degrade with increased scale, struggles to capture high-frequency details effectively.

output. However, limited research has contributed to explaining the reason and finding a proper frequency projection for input (Landgraf et al., 2022; Yüce et al., 2022).

In this paper, we aim to provide both a theoretical explanation and a proper solution to the inherent drawbacks of Fourier features embedding for INRs to prevent oversmoothness or noisy outputs. Firstly, a theoretical explanation is provided for the noisy output by examining the relationship between the eigenfunctions of MLPs with Fourier features and the Fourier series expansion of the target function. It is revealed by the analysis that high-frequency noise arises from finite sampling, indicating that high-frequency inputs accelerate the learning speed of a series of corresponding high-frequency targets, while unsampled frequencies establish a lower bound for the minimum loss.

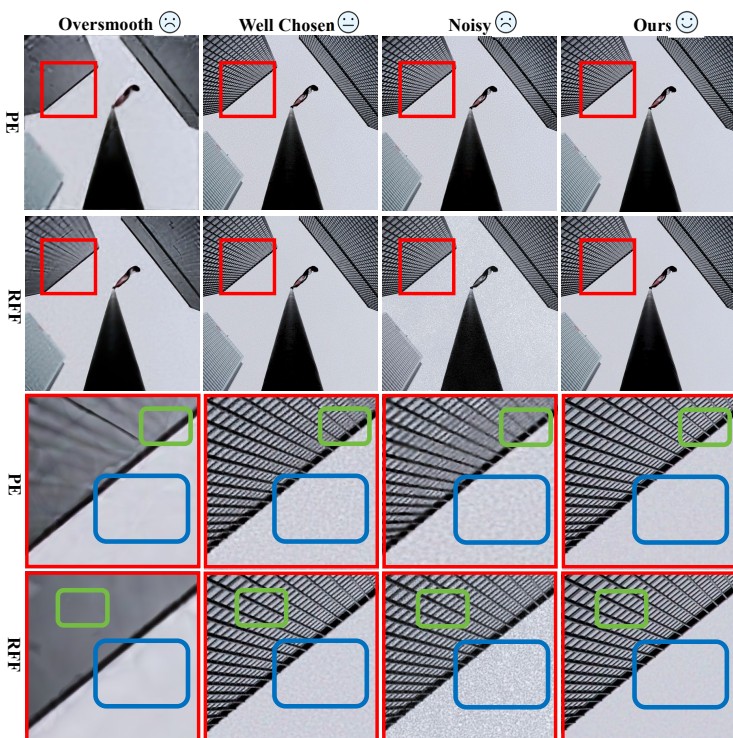

Figure 2: From the circled blue regions and green regions, it can be observed that even with well-chosen variance/scale, as experimented in Figure 1, the results are still unsatisfactory. However, using our proposed method, the noise is significantly alleviated while further enhancing the high-frequency details (Zoom in for a better view).

Inspired by the analysis of noisy output and the properties of Fourier series expansion, one approach to address this issue is to enable INRs to adaptively filter out unnecessary high-frequency components in low-frequency regions. Therefore, bias-free MLPs are employed, where bias-free means no biased terms are involved in any layer, functioning as an adaptive linear filter due to their scale-invariant property (Mohan et al., 2019) that ensures that the input pattern is maintained through each activation layer. Moreover, by viewing the learning rate of the proposed filter and INRs as a Pareto efficiency problem, a custom line-search algorithm is introduced to adjust the learning rate during training by solving an optimization problem and approximating a global minimum solution. By integrating these approaches, the performance in both low-frequency and high-frequency regions improved significantly, as shown in the comparison

in Figure 2. Finally, to evaluate the performance of the proposed method, we test it on various INRs tasks and compare it with state-of-the-art models, including BACON (Lindell et al., 2022) and SIREN (Sitzmann et al., 2020). The experimental results prove that our approach enables MLPs to capture finer details via Fourier Features while effectively reducing high-frequency noise without causing oversmoothness. To summarize, the followings are the main contributions of this work:

- From the Neural Tangent Kernel perspective, we provide a theoretical analysis of the noisy output issue caused by Fourier features embedding. This analysis further guides the design of our solution to this problem.
- We propose a method that applies a bias-free MLP as an adaptive linear filter to suppress unnecessary high frequencies. Additionally, a custom line-search algorithm is introduced to dynamically optimize the learning rate, achieving Pareto efficiency between the filter and INRs modules.
- To validate our approach, we conduct extensive experiments across a variety of tasks, including image regression, 3D shape regression, and Neural Radiance Field. These experiments demonstrate the effectiveness of our method in significantly reducing noisy outputs while avoiding the common issue of excessive smoothing, maintaining a balance between reducing noise and preserving high-frequency details.

## 2 RELATED WORKS

### 2.1 IMPLICIT NEURAL REPRESENTATIONS

Implicit Neural Representations are designed to learn a continuous representations of target functions by taking advantages of the approximation power of neural networks. Their inherent continuous property can beneficial in many cases like video compression (Chen et al., 2021a; Strümpler et al., 2022), 3D modeling (Park et al., 2019; Atzmon & Lipman, 2020; Michalkiewicz et al., 2019; Gropp et al., 2020; Sitzmann et al., 2019) and volume rendering (Pumarola et al., 2021; Barron et al., 2021; Martin-Brualla et al., 2021; Barron et al., 2023). However, simply employing MLPs may result in spectral bias, where oversmoothed outputs are generated due to the inherent tendency of MLPs to prioritize learning low-frequency components first. Consequently, many studies have focused on these drawbacks and explored various methods to address this issue. The most straightforward way to address this issue is by projecting the coordinates into the higher dimension (Tancik et al., 2020; Wang et al., 2021). However, these methods can lead to noisy outputs if there is a mismatch in the projection variance. To address this, Landgraf et al. (2022) propose dividing the Random Fourier Features into multiple levels of detail, allowing the MLPs to disregard unnecessary high-frequency components. Another type of approach to mitigating the spectral bias introduced by the ReLU activation function, as proposed by Sitzmann et al. (2020), Saragadam et al. (2023), and Shenouda et al. (2024), is to modify the activation function itself by using alternatives such as the Sine function, Wavelets, or a combination of ReLU with other functions. There are also efforts to modify network structures to mitigate spectral bias (Mujkanovic et al., 2024). Lindell et al. (2022) introduce a network design that treats MLPs as filters applied to the input of the next layer, known as Multiplicative Filter Networks (MFNs). Additionally, based on the discrete nature of signals like images and videos, grid-based approaches (e.g., Grid Tangent Kernel (Zhao et al., 2024), DINER (Xie et al., 2023), and Fourier Filter Bank (Wu et al., 2023)) have been proposed to address spectral bias, as the grid property allows for sharp changes in features, which facilitates learning fine details.

### 2.2 NEURAL TANGENT KERNEL

Deep neural networks are powerful across various domains but remain a black box that lacks interpretability. Therefore, many researchers have dived into explaining the mechanism of the neural networks in recent years. Lee et al. (2020) propose a Neural Network Gaussian Process (NNGP), modeling a two-layer neural network using a frozen first layer as the kernel, transforming it into kernel regression. Jacot et al. (2018); Arora et al. (2019b) introduce Neural Tangent Kernel (NTK) by linearizing the MLPs, extendable to multiple layers via induction, offering insights like spectral bias (Bietti & Mairal, 2019) and data distribution effects (Basri et al., 2020). Convolutional Neural Tangent Kernel (CNTK) (Arora et al., 2019b) generalizes these ideas to CNNs, enhancing researchers' understanding of different phenomena in deep learning (Tachella et al., 2021; Ulyanov et al., 2018; Cao & Gu, 2019; Advani et al., 2020).

## 3 PRELIMINARY OF FOURIER FEATURES

Fourier features are common embedding methods to alleviate spectral bias. As a type of embedding that maps inputs into the frequency domain, they can be expressed by the function $\gamma(\cdot) : X \in \mathbb{R}^d \to \mathbb{R}^N$, where $d$ is the input coordinate dimension and $N$ is the embedding dimension. The two most common types are Random Fourier Features (RFF) and Positional Encoding (PE), which can both be represented by a single formula with slight variations in their implementation.

**Definition 1** (**Fourier features**). **Fourier features** *can be generally defined as a function such that* $\gamma(\cdot) : X \in \mathbb{R}^d \to \mathbb{R}^N$

$$\gamma(\mathbf{x}) = [sin(2\pi \mathbf{b}_i^\top \mathbf{x}), cos(2\pi \mathbf{b}_i^\top \mathbf{x})]_{i \in [N]}, [N] = \{1, 2, 3, \cdots, N\}, \mathbf{b}_i \in \mathbb{R}^{d \times 1} \tag{1}$$

**Positional Encoding**: $\gamma(\mathbf{x}) = [sin(2\pi \sigma^{\frac{i}{n}\top} \mathbf{x}), cos(2\pi \sigma^{\frac{i}{n}\top} \mathbf{x})]_{i \in [N]}, [N] = \{1, 2, 3, \cdots, N\}$. *It applies log-linearly spaced frequencies for each dimension, with the scale $\sigma$ and size of embedding $N$ as hyperparameters, and includes only on-axis frequencies.*

**Random Fourier Features**: $\mathbf{b}_i \sim \mathcal{N}(0, \Sigma)$. *Typically, this is an isotropic Gaussian distribution, meaning that $\Sigma$ has only diagonal entries. Other distributions, such as the Uniform distribution, can also be used, though the Gaussian distribution remains the most common choice.*

## 4 THEORETICAL ANALYSIS OF FOURIER FEATURES

In this section, we examine why Fourier features can introduce high-frequency noise, from the persepctive of the Neural Tangent Kernel (NTK) derived from two-layer MLPs with a frozen second layer for simplicity as in Arora et al. (2019a). Our experiments in Figure 1 and Figure 2 confirm that the conclusion also stands in multi-layer MLPs. This analysis also helps explain why Positional Encoding might be more stable than Random Fourier Features in certain cases. By decomposing the target function as its Fourier series, we observe that MLPs primarily learn the given frequency components globally, leaving high-frequency components remain in smoother regions. Proofs for all theorems can be found in Appendix A.3.

The two following theorems, based on NTK (please check Appendix A.2 for the detailed formula), demonstrate that two-layer MLPs incorporated with Fourier Features essentially fit the target function by leveraging sampled frequencies and their combinations. If the sampled frequencies are integers, the unsampled frequencies impose a lower bound on the minimum achievable loss, meaning that finite sampling introduces noise primarily driven by these unsampled frequencies.

**Theorem 1.** *For a two-layer Multilayer-perceptrons (MLPs) denoted as $f(\mathbf{x}; \mathbf{W})$, where $\mathbf{x} \in \mathbb{R}^d$ as input and $\mathbf{W}$ as the parameters of the MLPs. Then the order-N approximation of eigenvectors of the Neural Tangent Kernel (Eq.5) when using Fourier features embedding, as defined in Def.1, to project the input to the frequency space can be presented as,*

$$k(\gamma(\mathbf{x}), \gamma(\mathbf{z})) = \sum_{i=1}^{N^\dagger} \lambda_i^2 cos(\mathbf{b}^* \mathbf{x}) cos(\mathbf{b}^* \mathbf{z}) + \sum_{i=1}^{N^\dagger} \lambda_i^2 sin(\mathbf{b}^* \mathbf{x}) sin(\mathbf{b}^* \mathbf{z}), \text{ where } N^\dagger \le 4Nk^m km^2 \tag{2}$$

*where*

$$\mathbf{b}^* \in \mathcal{L}_{Span\{b_j\}} \equiv \left\{ \mathbf{b}^* = \sum_{j=1}^n c_j \mathbf{b}_j \,\middle|\, \sum_{j=1}^\infty |c_j| < N + k^m km + m \right\} \tag{3}$$

*and $\lambda_i$s are eigenvalues for each eigenfunctions $sin(\mathbf{b}^* \mathbf{x})$ and $cos(\mathbf{b}^* \mathbf{x})$.*

**Theorem 2.** *For a d-dimensional target function $\mathbf{y}(\mathbf{x}) = \sum_{\mathbf{n} \in \mathbb{Z}^d} \hat{y}_\mathbf{n} e^{i\mathbf{n}^\top \mathbf{x}}$, where $\hat{y}_\mathbf{n}$ are the corresponding coefficients of the Fourier series expansion of the $\mathbf{y}(\mathbf{x})$. Given a pre-sampled frequency set $\mathbf{B}_n = \{\mathbf{b}_i \in \mathbb{Z}^d\}_{i \in [N]}$ and the $L_2$ loss function as $\phi(\mathbf{y}, f(\mathbf{x}; \mathbf{W})) = ||f(\mathbf{x}; \mathbf{W}) - \mathbf{y}||_2$. Let the projection of $\mathbf{y}(\mathbf{x})$ onto the spanned space of frequency set $\mathbf{B}_n$ be denoted by $\mathbf{y}_\mathbf{B}$ and the projection onto the orthogonal complement of this spanned space by $\mathbf{y}_\mathbf{B}^\dagger$ such that $\mathbf{y} = \mathbf{y}_\mathbf{B}^\dagger + \mathbf{y}_\mathbf{B}$. Then, with probability at least $1 - \delta$, for all $k = 0, 1, 2, \cdots$ (iteration numbers), the lower bound of the loss function can be represented as:*

$$||\mathbf{y}_\mathbf{B}^\dagger||_2 - \sqrt{\sum (1 - \eta \lambda_i)^{2k} \langle \mathbf{v}_i, \mathbf{y}_\mathbf{B} \rangle^2} \pm \epsilon \le \phi(\mathbf{y}, f(\mathbf{x}; \mathbf{W})) \tag{4}$$

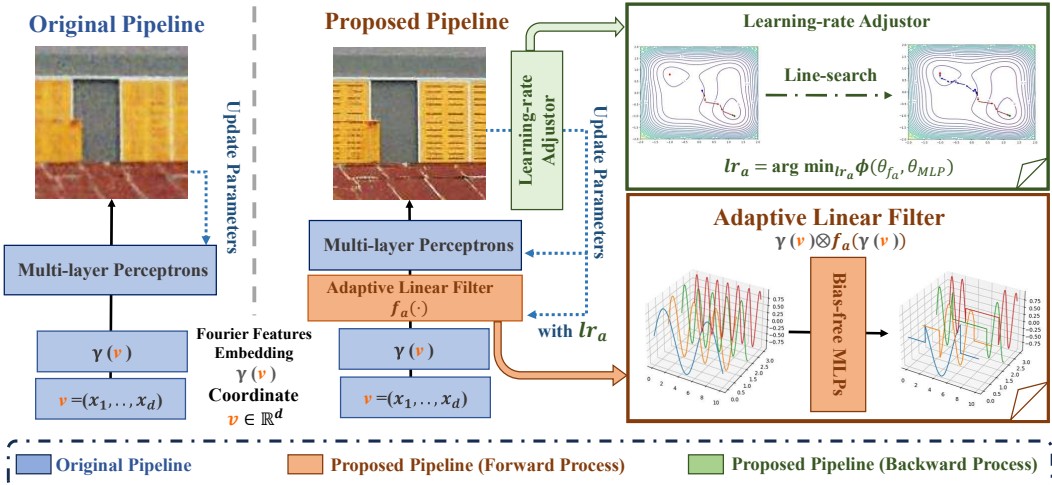

Figure 3: The pipeline of our method introduces two additional modules compared to the original approach. The first module, an adaptive linear filter, removes unnecessary frequency components at the pixel level, reducing high-frequency noise during regression. The second module dynamically adjusts the learning rate during training to optimize the approximated loss for the next step, achieving Pareto efficiency. Together, these modules result in cleaner and more detailed images.

To extend the analysis, we also examine the continuous frequencies sampled from $\mathbb{R}^d$. The next result demonstrates that the decay rate for integer frequencies close to the eigenfunctions is larger, aligning with and extending Theorem 2.

**Theorem 3.** *For a d-dimensional target function* $\mathbf{y}(\mathbf{x}) = \sum_{\mathbf{n} \in \mathbb{Z}^d} \hat{\mathbf{y}}_{\mathbf{n}} e^{i\mathbf{n}^\top \mathbf{x}}$, *where* $\hat{\mathbf{y}}_{\mathbf{n}}$ *are the corresponding coefficients of the Fourier series expansion of the* $\mathbf{y}(\mathbf{x})$. *Given a pre-sampled frequency set* $\mathbf{B}_n = \{\mathbf{b}_i \in \mathbb{R}^d\}_{i \in [N]}$ *and the* $L_2$ *loss function as* $\phi(\mathbf{y}, f(\mathbf{x}; \mathbf{W})) = ||f(\mathbf{x}; \mathbf{W}) - \mathbf{y}||_2$. *Then, for the frequency component* $\mathbf{n} \in \mathbb{Z}^d$, *and a sampled frequency* $\mathbf{b} \in \mathbf{B}_n$ *and its decomposition into the integer,* $\mathbf{b}_z$, *and residual part,* $\mathbf{b}_r \in [0, 1)$, *the decreasing rate of the loss function for specific frequency* $\mathbf{n}$ *from the target function using two-layers MLPs with second layer frozen is* $\mathcal{O}(\frac{1}{\prod_i [(\mathbf{n}+\mathbf{b}_z)+\mathbf{b}_r]_i} + \frac{1}{\prod_i [(\mathbf{n}-\mathbf{b}_z)-\mathbf{b}_r]_i})$.

The last lemma explains why Positional Encoding is more stable than Random Fourier Features in 2D case. Positional Encoding struggles with tilted high-frequency components, while Random Fourier Features, due to their high variance, mix high- and low-frequency signals, making it similarly difficult to capture low-frequency components. Intuitively, with Random Fourier Features, if the closest sampled frequency to a target's low-frequency component contains high-frequency elements, the noise gets introduced into the fitting result.

**Lemma 1.** *Considering two different Fourier features, Positional Encoding, and Random Fourier Features as in Def. 1. For two sampled frequencies using two embedding,* $\mathbf{b}_{pe}$ *from Positional Encoding and* $\mathbf{b}_{rff}$ *from Random Fourier Features, assume* $\mathbf{b}_{rff}$ *has two components with* $[\mathbf{b}_{rff}]_1 \gg [\mathbf{b}_{rff}]_2$, *and* $\mathbf{b}_{pe}$ *has only one non-zero component,* $[\mathbf{b}_{pe}]_2$, *equal to* $[\mathbf{b}_{rff}]_2$. *Let* $\mathbf{b}_z$ *be the closest integer frequency to* $\mathbf{b}_{rff}$ *and* $[\mathbf{b}_{rff}]_2 = [\mathbf{b}_z]_2$. *Then the decay rate of* $\mathbf{b}_z$ *for Positional Encoding,* $\mathbf{b}_{pe}$, *is equal to* $[\mathbf{b}_z]_2$ *for Random Fourier Features.*

## 5 METHODS

In this section, to tackle the noisy output, we present our solution grounded in the analysis of the cause. The proposed method has two main components: (i) an Adaptive Linear Filter that blocks irrelevant input frequencies during the forward pass, and (ii) a Learning-rate Adjustor that uses the line-search method during backpropagation to dynamically adjust the filter's learning rate. The full pipeline is illustrated in Figure 3.

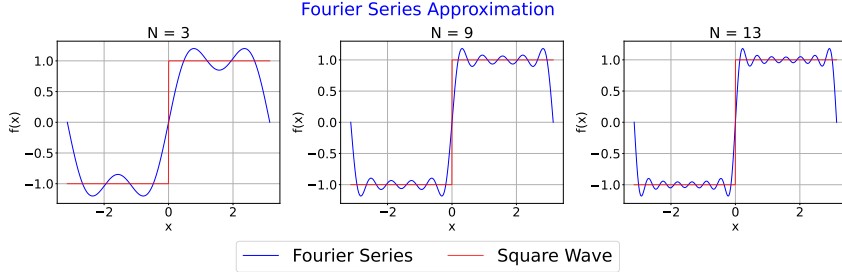

Figure 4: Finitely sampled Fourier series may introduce high-frequency noise in flat regions. If MLPs fail to capture sufficient frequencies, high-frequency noise persists in low-frequency regions.

## 5.1 BIAS-FREE MLPs AS ADAPTIVE LINEAR FILTER

Building on this analysis, MLPs can be viewed as a linear combination of eigenfunction frequencies, where MLPs utilize these frequencies as a prior to fit the Fourier series expansion of the target function. However, since the eigenfunctions' frequencies cannot represent all $m \in \mathbb{Z}^d$, noise may arise in the low-frequency regions. This is demonstrated in a 1-dimensional toy example (Figure 4) that with a limited number of frequency components, the fitted function struggles to suppress high-frequency components in low-frequency regions. Inspired by this observation, constraining the low-frequency regions to contain only low-frequency elements can significantly mitigate the issue.

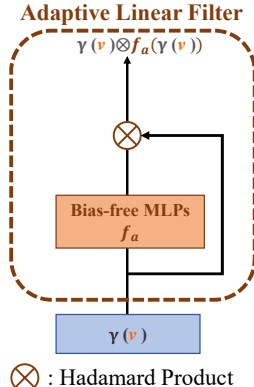

Figure 5: Illustration of the adaptive linear filter.

Therefore, we propose using bias-free MLPs as an adaptive band-limited coordinate-level linear filter for continuous representations as shown in Figure 5. Bias-free MLPs act as a linear filter that their output matches the size of the input Fourier features embedding and is then used to perform a coordinate-wise Hadamard product to filter the embedding. The bias-free network is chosen for its scale-invariance (Mohan et al., 2019), which preserves input frequency patterns when using ReLU activation. This ensures that if scaling the embedding by a constant, MLPs maintain the same amplitude and keep it at 0 for 0 inputs. Additionally, its local linearity enables the network to function as an adaptive linear filter, applying different linear terms to each coordinate, to selectively attenuate unnecessary components. Furthermore, This approach can also be extended to continuous-space tasks, such as 3D shape regression and inverse graphics, where the input of INRs is continuous rather than discrete, like image coordinates, benefiting from the continuity of MLPs. To verify the performance of this filter, we also visualized the filtered results in Appendix A.6. This visualization confirms that the proposed module effectively filters high-frequency inputs, preventing noisy outputs.

## 5.2 LINE-SEARCHED BASED OPTIMIZATION

During experiments with the adaptive filter, we observe that different initial learning rates for the adaptive linear filter and INRs led to varying performance outcomes as shown in Figure 6. This reflects a Pareto efficiency issue, where balancing the performance between the INRs and adaptive

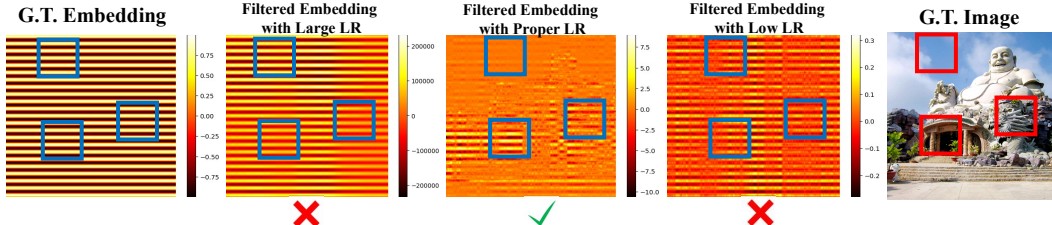

Figure 6: Both high- and low-learning rates result in unsatisfactory outcomes, where high-frequency components aren't suppressed in low-frequency regions (LR in this figure stands for learning rate).

linear filter is essential for optimal results. If the INRs learn significantly faster than the linear filter, the entire system may fall into local minima, where the adaptive linear filter fails to perform optimally. Conversely, if the adaptive linear filter's learning rate is too large, the input to the INR may fluctuate excessively, making it difficult for the INR to converge.

Inspired by Hao et al. (2021), we aim to optimize the learning rate of the adaptive linear filter. By optimizing the loss function $f(\theta_A, \theta_I)$ as $\phi(\alpha_A) = f(\theta_A, \theta_I)$ during training (where $\theta_A$ represents the parameters of the adaptive linear filter, $\theta_I$ represents the parameters of the INRs, $\alpha_A$ and $\alpha_I$ represent the learning rates for the adaptive filter and INR, respectively), we calculate the learning rate $\alpha_A$ for the adaptive linear filter at each iteration. By applying the Taylor expansion of the loss function, this optimization problem can be approximated as a linear optimization problem. Figure 7 provides an overview of how the proposed algorithm finds an optimal learning rate (derivation and algorithm are presented in Appendix A.5, Algorithm 3, and Algorithm 4).

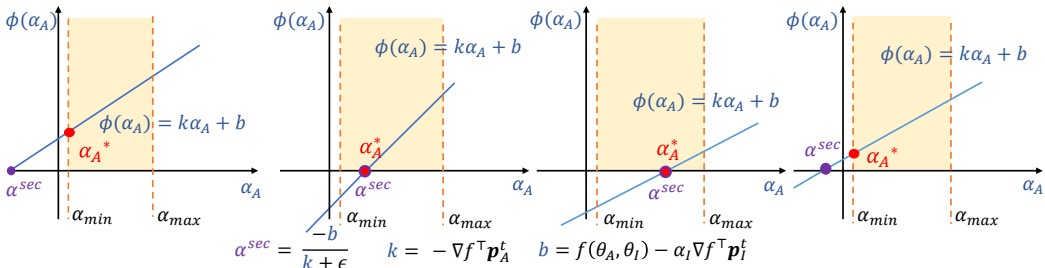

Figure 7: The blue line is the optimization target, while the orange lines indicate the predefined learning rate bounds, denoted as $\alpha_{\min}$ and $\alpha_{\max}$. $\mathbf{p}_A^t$ and $\mathbf{p}_I^t$ are the update directions for the filter and INRs, respectively. $\alpha^*$ is the optimal value and $\epsilon$ is a constant for robustness, usually set to $1 \times 10^{-6}$.

# 6 EXPERIMENTS

To validate the proposed method, we test it across various tasks, including image regression, 3D shape regression, and inverse graphics. All experiments are performed on a single RTX 4090 GPU, using an adaptive linear filter with 3 layers, each with the same width as the number of channels in the Fourier features embedding.

## 6.1 IMAGE REGRESSION

**Setup and Implementation Details:** Following prior research, we use the validation split of the DIV2K dataset (Agustsson & Timofte, 2017), which consists of 100 natural images at 2K resolution, featuring a diverse range of content. The experiments are conducted under a resolution of $256 \times 256$. The models are trained using the mean squared error (MSE) loss. We compared our proposed method with several baselines: Multi-Layer Perceptron (MLP) with Positional Encoding (96 sampled frequencies per dimension), MLP with Random Fourier Features (384 sampled frequencies), SIREN (Sitzmann et al., 2020), and BACON (Lindell et al., 2022). Each model is trained for 10,000 iterations to ensure convergence, with the learning rate $1 \times 10^{-3}$. For the custom line-search algorithm, we set the maximum learning rate as $1 \times 10^{-2}$, with a minimum of 0. To provide a more comprehensive comparison, we evaluate the performance on three metrics: PSNR, SSIM, and LPIPS (Zhang et al., 2018)

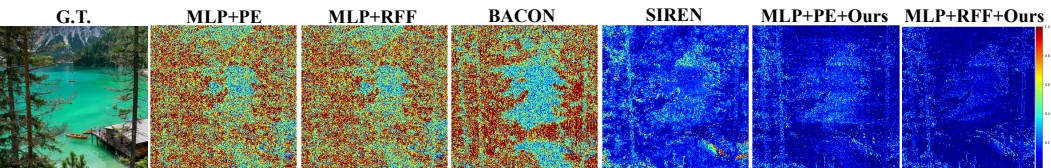

Figure 8: The absolute error map between the ground truth image and the fitted result. The closer to red, the larger the error; the closer to blue, the smaller the error.

**Experiment Results:** As shown in Figure 8, our method outperforms not only baseline MLPs with Fourier features but also the other two models when visualizing using the error map. In Table 1, both MLP+RFF+Ours and MLP+PE+Ours demonstrate superior performance in PSNR, SSIM, and LPIPS. Specifically, MLP+RFF+Ours achieves the highest PSNR of 53.38 and an SSIM of 0.9967. MLP+PE+Ours excels in noise reduction with a low LPIPS of 0.0002. Overall, our method significantly improves image regression in detail reconstruction.

Table 1: Performance comparison of image regression tasks across different methods. We highlight the best results in bold and underline the second-best results.

|  | MLP+PE | MLP+RFF | BACON | SIREN | MLP+PE+Ours | MLP+RFF+Ours |
|---|---|---|---|---|---|---|
| PSNR↑ | 36.49 | 36.86 | 36.36 | 47.78 | 49.14 | **53.38** |
| SSIM↑ | 0.9594 | 0.9604 | 0.9750 | 0.9944 | **0.9967** | 0.9908 |
| LPIPS↓ | 0.0173 | 0.0164 | 0.0012 | 0.0005 | **0.0002** | 0.0075 |

## 6.2 3D-SHAPE REGRESSION

**Setup and Implementation Details:** We evaluate our method on the Signed-Distance-Function (SDF) regression task, aiming to learn a function that maps 3D coordinates to their signed distance values. Positive values indicate points outside an object, and negative values are inside. The objective is precise 3D shape reconstruction. We follow the experimental setup from Lindell et al. (2022), training each model for 200,000 iterations with a learning rate starting at $1 \times 10^{-3}$. The learning rate for line-search was capped at $1 \times 10^{-3}$. Performance is evaluated using Chamfer Distance and IOU (Intersection over Union), evaluating four Stanford 3D Scanning Repository scenes [1]: Armadillo, Dragon, Lucy, and Thai, each with 10,000 sampled points. To calculate the IOU score, we evaluate the intersection and union of occupancy values between the ground truth and predicted meshes on a $128^3$ grid of points centered around the object following the idea from BACON (Lindell et al., 2022). And using the ground truth sampled 10000 points from the object's surface for the Chamfer Distance. Our comparisons include the following baselines: Multi-Layer Perceptron (MLP) with Fourier Features (including Random Fourier Features and Positional Encoding with 64 sampled frequencies per dimension), SIREN(Sitzmann et al., 2020), and BACON.

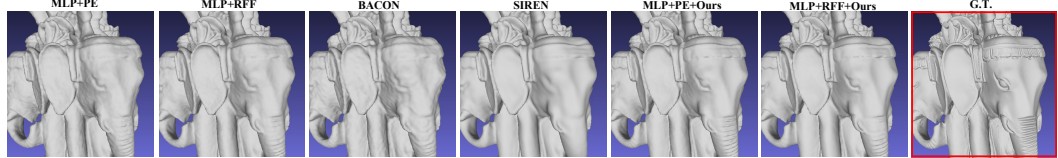

Figure 9: Visualization of the 3D shape regression task results (Zoom in for a better view).

**Experiment Results:** From quantification results shown in the Table 2, Fourier Features+our method achieves the lowest Chamfer Distance and highest IOU score, demonstrating superior accuracy in shape reconstruction. Illustrations of results can be found at Figure 9 and Appendix A.7.1, where it can be found that the proposed method, to some extent, smoothed the surface while reconstructing more details compared with other baselines.

## 6.3 NEURAL RADIANCE FIELD

**Setup and Implementation Details:** This section discusses fitting 3D scenes using Neural Radiance Fields (NeRF), aiming at reconstructing scenes by predicting color and density based on 3D coordinates and viewing direction. The models use an MSE loss and are trained for 1,000,000 iterations with an initial learning rate of $5 \times 10^{-4}$. Performance is evaluated with PSNR, SSIM, and LPIPS.

|  | BACON | MLP+PE | MLP+PE+Ours |
|---|---|---|---|
| PSNR↑ | 28.14 | 30.87 | **31.37** |
| SSIM↑ | 0.9291 | 0.9486 | **0.9544** |
| LPIPS↓ | 0.0436 | 0.0291 | **0.0241** |

Table 3: The quantification result of NeRF task for baselines.

---

[1]http://graphics.stanford.edu/data/3Dscanrep/

Table 2: 3D shape regression metrics across baseline methods. We highlight the best results in bold and underline the second-best results.

| Metric | MLP+PE | MLP+RFF | BACON | SIREN | MLP+PE+Ours | MLP+RFF+Ours |
|---|---|---|---|---|---|---|
| Chamfer Distance ($\downarrow$) | 1.8413e-06 | 1.8525e-06 | 1.9535e-06 | 1.8313e-06 | **1.7919e-06** | 1.7947e-06 |
| IOU ($\uparrow$) | 0.96189 | 0.96226 | 0.96168 | 0.96217 | 0.96245 | **0.96247** |

We applied line-search (from $1 \times 10^{-3}$ to 0) to minimize overfitting, evaluating at the NeRF Blender dataset (Martin-Brualla et al., 2021), which consists of diverse synthetic scenes. Training used cropped $400\times400$ images with a white background for consistency. Our comparisons involved a baseline MLP with Positional Encoding (64 sampled frequencies per dimension), BACON (Lindell et al., 2022), and our full method. RFF and SIREN was excluded due to their instability in higher-dimensional space and making it less suitable for this task.

**Experiment Results:** The results in Table 3 show that our proposed method surpasses both the vanilla NeRF and the BACON-based NeRF. As depicted in Figure 10, our approach enables NeRF to capture finer details, such as the caterpillar tracks and the Phillips head on the Lego model.

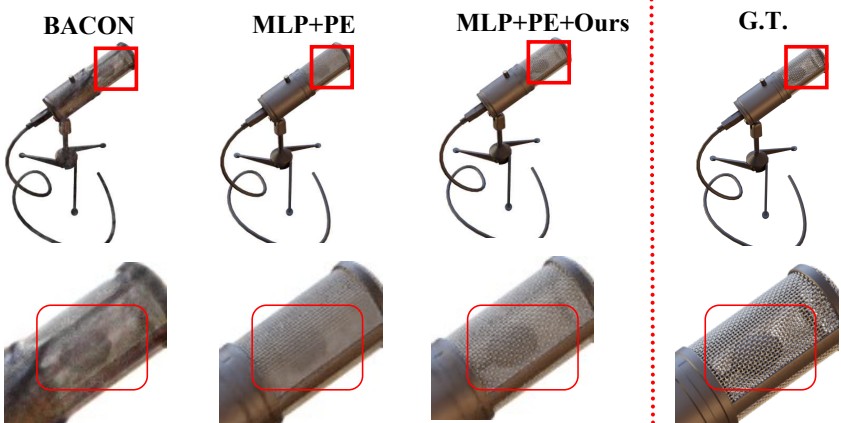

Figure 10: Comparison of visual results of NeRF task with baselines (Zoom in for a better view). More visualization results are available at Appendix.A.7.2

## 6.4 ABLATION STUDY

In this section, we evaluate the line-search method's performance, finding it achieves a better result compared to using only the adaptive linear filter, especially for image regression tasks. This ablation study confirms that the proposed line-search algorithm is more effective at finding an optimal minimum. The results are shown in Table 4.

| | Image Regression | | | 3D Shape Regression | | NeRF | | |
|---|---|---|---|---|---|---|---|---|
| | PSNR$\uparrow$ | SSIM$\uparrow$ | LPIPS$\downarrow$ | Chamfer Distance$\downarrow$ | IOU$\uparrow$ | PSNR$\uparrow$ | SSIM$\uparrow$ | LPIPS$\downarrow$ |
| MLP + PE + Ours w/o L | 45.02 | 0.9860 | 0.0080 | 1.8058e-6 | 0.96240 | 31.24 | 0.9534 | 0.0254 |
| MLP + PE + Ours w/L | **49.14** | **0.9967** | **0.00019** | **1.7919e-6** | **0.96245** | **31.37** | **0.9544** | **0.0241** |
| MLP + RFF + Ours w/o L | 50.36 | 0.9898 | 0.0076 | 1.8159e-6 | 0.96234 | ╲ | ╲ | ╲ |
| MLP + RFF + Ours w/L | **53.38** | **0.9908** | **0.0075** | **1.7947e-6** | **0.96247** | ╲ | ╲ | ╲ |

Table 4: Performance comparison of various methods for Image Regression, 3D Shape Regression, and NeRF tasks. "w/o" stands for "without," "w/" stands for "with," and "L" refers to our custom line-search algorithm.

### 6.4.1 CONVERGENCE OF MODIFIED LINE-SEARCH ALGORITHM

To address concerns about potential divergence, we validate the convergence of the modified line-search algorithm through experiments on the DIV2K validation split, including both RFF and PE. The results demonstrate that the algorithm converges for both embeddings. As illustrated in Figure 11, the learning rates of the adaptive linear filter for both embeddings consistently decrease

throughout training and ultimately converge to 0. This steady reduction in learning rates confirms the stability and convergence of the algorithm by the end of the training process.

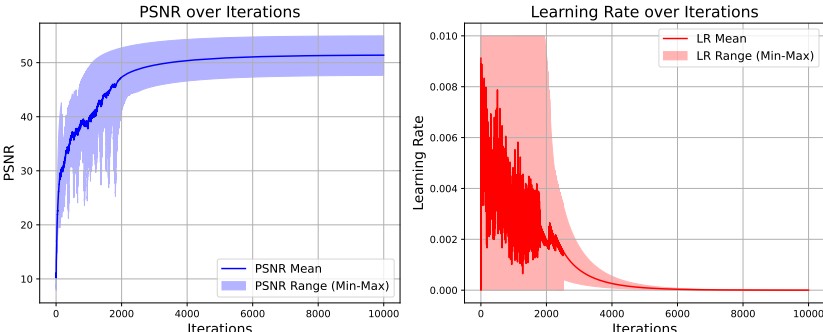

Figure 11: We demonstrate the convergence of the modified line-search algorithm through image regression experiments for RFF and PE. During training, both the PSNR and the learning rate consistently converge, confirming the effectiveness of our proposed line-search-based approach.

### 6.4.2 VARYING VARIANCE

We also evaluate the impact of varying variance on the same image regression task as in Figure 1 using our proposed method. As shown in Figure 12, unlike the results presented in Figure 1, performance remains stable even with high sampling variance when our method is applied. This highlights the robustness of our approach under high sampling variance.

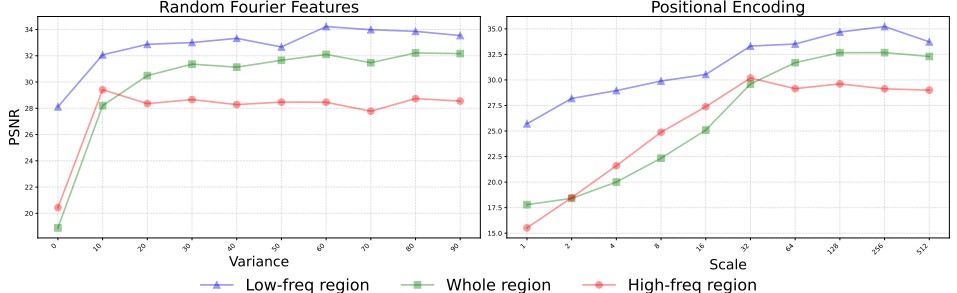

Figure 12: We evaluated whether our proposed method could mitigate the high-frequency phenomenon associated with varying variances in two Fourier features embedding methods. The results indicate that our method successfully prevents model degradation even under conditions of high variance for RFF, where traditional embeddings fail to perform effectively.

## 7 CONCLUSION AND LIMITATIONS

Building on insights from the Neural Tangent Kernel (NTK), we analyze the high-frequency noise in Fourier Features, which arises due to limited frequency sampling. This understanding motivates the development of our proposed method, which incorporates a line-search algorithm to achieve a Pareto-efficient balance between frequency learning and noise reduction. By applying our method to a range of tasks, including image regression, 3D shape regression, and Neural Radiance Fields (NeRF), we consistently outperform baseline models. Our approach excels at capturing high-frequency details while effectively mitigating noise, leading to more accurate reconstructions. The method demonstrates robust performance in both low- and high-frequency regions, ensuring more precise and stable outputs in complex tasks.

**Limitations:** Despite the improvements, our method does not completely resolve finite sampling issues from the root. Additionally, while the line-search algorithm enhances the performance of the adaptive linear filter, it may lead to slower convergence and occasional instability during the early stage of training. Addressing these challenges is part of our future work.

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

# A APPENDIX

## A.1 DEFINITION OF HIGH-DIMENSIONAL FOURIER SERIES

For a d-dimensional periodic function $f(\mathbf{x})$ with input $\mathbf{x} = [x_1, x_2, \cdots, x_d]^\top$ be a $2\pi$ period function with respect to each components. Then the function $f(\mathbf{x})$ can be expanded as:

$$f(\mathbf{x}) = \sum_{\mathbf{m} \in \mathbb{Z}^d} \hat{f}_m \mathbf{e}^{i\mathbf{m}^\top \mathbf{x}}$$

where $\hat{f}_m$ is the coefficient of different frequency component.

## A.2 NEURAL TANGENT KERNEL

The Neural Tangent Kernel (NTK), a prominent tool for neural network analysis, has attracted considerable attention since its introduction. To simplify the analysis, this section will focus specifically on the NTK for two-layer MLPs, as the subsequent analysis also relies on the two-layer assumption. The two-layer MLP, $f(\mathbf{x}; \mathbf{w})$, with activation function $\sigma(\cdot)$ and input $\mathbf{x} \in \mathbb{R}^d$, can be expressed as follows:

$$f(\mathbf{x}; \mathbf{w}) = \frac{1}{\sqrt{m}} \sum_{r=1}^{m} a_r \sigma(\mathbf{w}_r^\top \mathbf{x} + \mathbf{b}_r)$$

where m is the width of the layer and $\|\mathbf{x}\| = 1$ (also can be written as $\mathbf{x} \in \mathbb{S}^{d-1}$, where $\mathbb{S}^{d-1} \equiv \{\mathbf{x} \in \mathbb{R}^d : \|\mathbf{x}\| = 1\}$). The term $\frac{1}{\sqrt{m}}$ is used to assist the analysis of the network. Based on this MLP, the kernel is defined as the following:

$$k(\mathbf{x_i}, \mathbf{x_j}) = \mathbb{E}_{\mathbf{w} \sim \mathcal{I}} \left\{ \left\langle \frac{\partial f(\mathbf{x_i}; \mathbf{w})}{\partial \mathbf{w}}, \frac{\partial f(\mathbf{x_j}; \mathbf{w})}{\partial \mathbf{w}} \right\rangle \right\}$$

This formula enables the exact expression of the NTK to better analyze the behavior and dynamics of MLP. For a two-layer MLP with a rectified linear unit (ReLU) activation function where only the first layer weights are trained and the second layer is frozen, the NTK of this network can be written as the following (Xie et al., 2017):

$$k(\mathbf{x_i}, \mathbf{x_j}) = \frac{1}{4\pi}(\langle \mathbf{x_i}, \mathbf{x_j} \rangle + 1)(\pi - arccos(\langle \mathbf{x_i}, \mathbf{x_j} \rangle)) \tag{5}$$

This expression can help us to determine the eigenfunction and eigenvalue of kernel and therefore provide a more insightful analysis of the network.

## A.3 PROOF OF THEOREMS

### A.3.1 PRELIMINARY LEMMAS

**Lemma 2.** *Let $\{\mathbf{b}_i^{(1)} \in \mathbb{R}^d\}_{i \in [N]}$ and $\{\mathbf{b}_j^{(2)} \in \mathbb{R}^d\}_{j \in [M]}$ be two sets of frequency vectors and N and M are integers that represent the size for each set, $\mathbf{x} \in \mathbb{R}^d$ is the coordinates in d-dimensional space. Then,*

$$\left( \sum_{i=1}^{N} c_i^{(1)} cos(\mathbf{b}_i^{(1)\top} \mathbf{x}) \right) \left( \sum_{j=1}^{M} c_j^{(2)} cos(\mathbf{b}_j^{(2)\top} \mathbf{x}) \right) = \left( \sum_{k=1}^{T} c_k^* cos(\mathbf{b}_k^{*\top} \mathbf{x}) \right), where\, T \leq 2NM$$

$$\tag{6}$$

*where,*

$$\mathbf{b}^* \in \left\{ \mathbf{b}^* = \mathbf{b}_i^{(1)} \pm \mathbf{b}_j^{(2)} \Big| i \in [N],\, j \in [M] \right\} \tag{7}$$

*Proof.*

$$\left( \sum_{i=1}^{N} c_i^{(1)} cos(\mathbf{b}_i^{(1)\top} \mathbf{x}) \right) \left( \sum_{j=1}^{M} c_j^{(2)} cos(\mathbf{b}_j^{(2)\top} \mathbf{x}) \right) \tag{8}$$

$$= \left( \sum_{i=1}^{N} \sum_{j=1}^{M} c_i^{(1)} c_j^{(2)} cos(\mathbf{b}_i^{(1)\top} \mathbf{x}) cos(\mathbf{b}_j^{(2)\top} \mathbf{x}) \right) \tag{9}$$

$$= \left( \sum_{i=1}^{N} \sum_{j=1}^{M} \frac{1}{2} c_i^{(1)} c_j^{(2)} \left( cos((\mathbf{b}_i^{(1)} + \mathbf{b}_j^{(2)})^{\top} \mathbf{x}) + cos((\mathbf{b}_i^{(1)} - \mathbf{b}_j^{(2)})^{\top} \mathbf{x}) \right) \right) \tag{10}$$

$$= \sum_{k=1}^{T} c_k^* cos(\mathbf{b}_k^{*\top} \mathbf{x}), where\, T \leq 2NM \tag{11}$$

$$\square$$

**Lemma 3.** *Let $\{\mathbf{b}_i \in \mathbb{R}^d\}_{i \in [n]}$ be a set of frequency vectors and N is an integer that represents the size, $\mathbf{x} \in \mathbb{R}^d$ is the coordinates in d-dimensional space. Then,*

$$\left( \sum_{i=1}^{n} cos(\mathbf{b}_i^{\top} \mathbf{x}) \right)^k = \left( \sum_{k=1}^{N} cos(\mathbf{b}_k^{*\top} \mathbf{x}) \right), where\, N \leq k^n nk \tag{12}$$

*where,*

$$\mathbf{b}^* \in \left\{ \mathbf{b}^* = \sum_{i}^{n} c_i \mathbf{b_i} \middle| c_i \in \mathbb{Z}, \sum_{i}^{n} |c_i| \leq k \right\} \tag{13}$$

*Proof.* **Proof by induction:**
**when k=1**
**This is a special case proved by Lemma 2.**
**Assuming the claim of this Lemma is true for k=m, then when k=m+1**

$$\left( \sum_{i=1}^{n} cos(\mathbf{b}_i^{\top} \mathbf{x}) \right)^{m+1} \tag{14}$$

$$= \left( \sum_{i=1}^{n} cos(\mathbf{b}_i^{\top} \mathbf{x}) \right)^m \left( \sum_{i=1}^{n} cos(\mathbf{b}_i^{\top} \mathbf{x}) \right) \tag{15}$$

$$\tag{16}$$

**By the assumption on k=m**

$$= \left( \sum_{k=1}^{n'} cos(\mathbf{b}_k^{\dagger\top} \mathbf{x}) \right) \left( \sum_{i=1}^{n} cos(\mathbf{b}_i^{\top} \mathbf{x}) \right), where\, n' \leq m^n nm \tag{17}$$

$$\textbf{where } \mathbf{b}^\dagger \in \left\{ \mathbf{b}^\dagger = \sum_{i}^{n} c_i \mathbf{b_i} \middle| c_i \in \mathbb{Z}, \sum_{i}^{n} |c_i| \leq m \right\} \tag{18}$$

$$\tag{19}$$

**By Lemma 2**

$$= \left( \sum_{k=1}^{N} cos(\mathbf{b}_k^{*\top} \mathbf{x}) \right), where \, N \leq (m+1)^n n(m+1) \tag{20}$$

$$\textbf{where } \mathbf{b}_k^{*\top} \in \left\{ \mathbf{b}^* = \sum_i^n c_i \mathbf{b_i} \pm \mathbf{b_j} \middle| c_i \in \mathbb{Z}, \sum_i^n |c_i| \leq m, \forall i, j \right\} \tag{21}$$

$$\Rightarrow \mathbf{b}_k^{*\top} \in \left\{ \mathbf{b}^* = \sum_i^n c_i^* \mathbf{b_i} \middle| c_i \in \mathbb{Z}, \sum_i^n |c_i| \leq m+1 \right\} \tag{22}$$

$\square$

**Lemma 4.** *Given a pre-sampled frequency set* $\mathbf{B}_n = \{\mathbf{b}_i \in \mathbb{N}^d\}_{i \in [N]}$ *and the Fourier features projection,* $\gamma(\cdot)$, *as* $\gamma(\mathbf{x}) = [sin(2\pi \mathbf{b}_i^\top \mathbf{x}), cos(2\pi \mathbf{b}_i^\top \mathbf{x})]_{i \in [N]}, [N] = 1, 2, 3, \cdots, N$. *Then,* $\gamma(\mathbf{x})^\top \gamma(\mathbf{z}) = sum(\gamma(\mathbf{x} - \mathbf{z}))$.

*Proof.*

$$\gamma(\mathbf{x})^\top \gamma(\mathbf{z}) = \sum_{i=1}^{N} cos(2\pi \mathbf{b}_i^\top \mathbf{x}) cos(2\pi \mathbf{b}_i^\top \mathbf{z}) + sin(2\pi \mathbf{b}_i^\top \mathbf{x}) sin(2\pi \mathbf{b}_i^\top \mathbf{z}) \tag{23}$$

$$= \sum_{i=1}^{N} cos(2\pi \mathbf{b}_i^\top (\mathbf{x} - \mathbf{z})) = sum(\gamma(\mathbf{x} - \mathbf{z})) \tag{24}$$

$\square$

**Theorem 1.** *For a two-layer Multilayer-perceptrons (MLPs) denoted as* $f(\mathbf{x}; \mathbf{W})$, *where* $\mathbf{x} \in \mathbb{R}^d$ *as input and* $\mathbf{W}$ *as the parameters of the MLPs. Then the order-N approximation of eigenvectors of the Neural Tangent Kernel (Eq.5) when using Fourier features embedding, as defined in Def.1, to project the input to the frequency space can be presented as,*

$$k(\gamma(\mathbf{x}), \gamma(\mathbf{z})) = \sum_{i=1}^{N^\dagger} \lambda_i^2 cos(\mathbf{b}^* \mathbf{x}) cos(\mathbf{b}^* \mathbf{z}) + \sum_{i=1}^{N^\dagger} \lambda_i^2 sin(\mathbf{b}^* \mathbf{x}) sin(\mathbf{b}^* \mathbf{z}), where \, N^\dagger \leq 4Nk^m km^2 \tag{25}$$

*where*

$$\mathbf{b}^* \in \mathcal{L}_{Span\{b_j\}} \equiv \left\{ \mathbf{b}^* = \sum_{j=1}^{n} c_j \mathbf{b}_j \middle| \sum_{j=1}^{\infty} |c_j| < N + k^m km + m \right\} \tag{26}$$

*and* $\lambda_i s$ *are eigenvalues for each eigenfunctions* $sin(\mathbf{b}^* \mathbf{x})$ *and* $cos(\mathbf{b}^* \mathbf{x})$.

*Proof.* By Xie et al. (2017), the two-layer MLP's NTK has the form as the following:

$$k(x, z) = \frac{\langle \mathbf{x}, \mathbf{z} \rangle (\pi - arccos(\langle \mathbf{x}, \mathbf{z} \rangle))}{2\pi}$$

If we use Fourier features mapping, $\gamma(\mathbf{x})$, before inputting to the Neural Network with a randomly sampled frequency set $\{\mathbf{b}_i\}_{i=1}^{m}$.

By the Lemma 4, in order to ensure that the vector dot product still be a valid dot product in $S^{d-1}$, the dot product of two embedded input can be written as $\gamma(\mathbf{x})^\top \gamma(\mathbf{z}) = \frac{1}{||\gamma(\mathbf{x})|| ||\gamma(\mathbf{z})||} \sum_{i=1}^{m} cos(2\pi \mathbf{b}_i(\mathbf{z} - \mathbf{x}))$ to make sure the dot product is bounded by 1.

$$k(\gamma(\mathbf{x}), \gamma(\mathbf{z})) = \frac{\langle \gamma(\mathbf{x}), \gamma(\mathbf{z}) \rangle (\pi - arccos(\langle \gamma(\mathbf{x}), \gamma(\mathbf{z}) \rangle))}{2\pi} \tag{27}$$

**Denoting** $||\gamma(\mathbf{x})|| ||\gamma(\mathbf{z})||$ **as** $\aleph$ $\tag{28}$

$$= \frac{\sum_{i=1}^m cos(2\pi \mathbf{b}_i(\mathbf{z} - \mathbf{x}))(\pi - arccos(\frac{1}{\aleph} \sum_{i=1}^m cos(2\pi \mathbf{b}_i(\mathbf{z} - \mathbf{x}))))}{2\pi\aleph} \tag{29}$$

**By N-order approximation Taylor Expansion of** $arccos(\cdot)$ $\tag{30}$

$$= \frac{\sum_{i=1}^m cos(2\pi \mathbf{b}_i(\mathbf{z} - \mathbf{x}))(\frac{\pi}{2} + \sum_{k=1}^N \frac{(2n)!}{2^{2n}(n!)^2}(\sum_{i=1}^m \frac{1}{\aleph} cos(2\pi \mathbf{b}_i(\mathbf{z} - \mathbf{x})))^k)}{2\pi\aleph} \tag{31}$$

**By Lemma 4** $\tag{32}$

$$= \frac{\sum_{i=1}^m cos(2\pi \mathbf{b}_i(\mathbf{z} - \mathbf{x}))(\frac{\pi}{2} + \sum_{k=1}^N \sum_{i=1}^M \beta_i^* cos(2\pi \mathbf{b}_i^*(\mathbf{z} - \mathbf{x})))}{2\pi\aleph}, \text{where } \mathbf{M} \le k^m km \tag{33}$$

$$where\ \mathbf{b}_i^* \in \left\{ \mathbf{b}^* = \sum_i^m c_i \mathbf{b_i} \middle| c_i \in \mathbb{Z}, \sum_i^n |c_i| \le k \right\} \tag{34}$$

$$= \frac{\sum_{i=1}^m cos(2\pi \mathbf{b}_i(\mathbf{z} - \mathbf{x}))(\frac{\pi}{2} + \sum_{i=1}^{N^*} \beta_i^* cos(2\pi \mathbf{b}_i^*(\mathbf{z} - \mathbf{x})))}{2\pi\aleph}, \text{where } N^* \le 2NM \tag{35}$$

$$where\ \mathbf{b}_i^* \in \left\{ \mathbf{b}^* = \sum_i^m c_i \mathbf{b_i} \middle| c_i \in \mathbb{Z}, \sum_i^n |c_i| \le N + M \right\} \tag{36}$$

$$= \frac{\frac{\pi}{2} \sum_{i=1}^m cos(2\pi \mathbf{b}_i(\mathbf{z} - \mathbf{x})) + \sum_{i=1}^m cos(2\pi \mathbf{b}_i(\mathbf{z} - \mathbf{x})) \sum_{i=1}^{N^*} \beta_i^* cos(2\pi \mathbf{b}_i^*(\mathbf{z} - \mathbf{x}))}{2\pi\aleph} \tag{37}$$

**By Lemma 3** $\tag{38}$

$$= \frac{\frac{\pi}{2} \sum_{i=1}^m cos(2\pi \mathbf{b}_i(\mathbf{z} - \mathbf{x})) + \sum_{i=1}^{N^\dagger} \beta_i^\dagger cos(2\pi \mathbf{b}_i^\dagger(\mathbf{z} - \mathbf{x}))}{2\pi\aleph}, \text{where } N^\dagger \le 2mN^* \tag{39}$$

$$where\ \mathbf{b}_i^\dagger \in \left\{ \mathbf{b}^\dagger = \sum_i^m c_i \mathbf{b_i} \middle| c_i \in \mathbb{Z}, \sum_i^n |c_i| \le N + M + m \right\} \tag{40}$$

$$= \frac{1}{4\aleph} \sum_{i=1}^m cos(2\pi \mathbf{b}_i(\mathbf{z} - \mathbf{x})) + \frac{1}{2\pi\aleph} \sum_{i=1}^{N^\dagger} \beta_i^\dagger cos(2\pi \mathbf{b}_i^\dagger(\mathbf{z} - \mathbf{x}))), \text{where } N^\dagger \le 4Nk^m km^2 \tag{41}$$

$$\tag{42}$$

Furthermore, to do the eigendecomposition, we need further to split this into the product of two orthogonal functions by $cos(a - b) = cos(a)cos(b) + sin(a)sin(b)$

$$= \frac{1}{4\aleph} \sum_{i=1}^m cos(2\pi \mathbf{b}_i\mathbf{x}))cos(2\pi \mathbf{b}_i\mathbf{z})) + sin(2\pi \mathbf{b}_i\mathbf{x})sin(2\pi \mathbf{b}_i\mathbf{z}) \tag{43}$$

$$+ \frac{1}{2\pi\aleph} \sum_{i=1}^{N^\dagger} \beta_i^\dagger cos(2\pi \mathbf{b}_i^\dagger\mathbf{x}))cos(2\pi \mathbf{b}_i^\dagger\mathbf{z})) + sin(2\pi \mathbf{b}_i^\dagger\mathbf{x})sin(2\pi \mathbf{b}_i^\dagger\mathbf{z}) \tag{44}$$

$\square$

**Theorem 2.** *For a d-dimensional target function* $\mathbf{y}(\mathbf{x}) = \sum_{\mathbf{n} \in \mathbb{Z}^d} \hat{y}_{\mathbf{n}} e^{i\mathbf{n}^\top \mathbf{x}}$*, where* $\hat{y}_{\mathbf{n}}$ *are the corresponding coefficients of the Fourier series expansion of the* $\mathbf{y}(\mathbf{x})$*. Given a pre-sampled frequency set* $\mathbf{B}_n = \{\mathbf{b}_i \in \mathbb{Z}^d\}_{i \in [N]}$ *and the* $L_2$ *loss function as* $\phi(\mathbf{y}, f(\mathbf{x}; \mathbf{W})) = ||f(\mathbf{x}; \mathbf{W}) - \mathbf{y}||_2$*. Let the projection of* $\mathbf{y}(\mathbf{x})$ *onto the spanned space of frequency set* $\mathbf{B}_n$ *be denoted by* $\mathbf{y_B}$ *and the projection onto the orthogonal complement of this spanned space by* $\mathbf{y_B^\dagger}$ *such that* $\mathbf{y} = \mathbf{y_B^\dagger} + \mathbf{y_B}$*. Then, with probability at least* $1 - \delta$*, for all* $k = 0, 1, 2, \cdots$ *(iteration numbers), the lower bound of the loss*

*function can be represented as:*

$$||\mathbf{y}_{\mathbf{B}}^{\dagger}||_2 - \sqrt{\sum (1 - \eta\lambda_i)^{2k}\langle \mathbf{v}_i, \mathbf{y_B}\rangle^2} \pm \epsilon \leq \phi(\mathbf{y}, f(\mathbf{x}; \mathbf{W})) \tag{45}$$

*Proof.* Given $\mathbf{B}_n = \{\mathbf{b}_i \in \mathbb{N}^d\}_{i \in [N]}$, this can be spanned as a subspace (by Theorem 1), $\left\{cos(2\pi\mathbf{b}^{\dagger}\mathbf{x}), sin(2\pi\mathbf{b}^{\dagger}\mathbf{x})|\mathbf{b}_i^{\dagger} \in \left\{\mathbf{b}^{\dagger} = \sum_i^m c_i\mathbf{b_i}\big| c_i \in \mathbb{Z}, \sum_i^n |c_i| \leq N^{\dagger}\right\}\right\}$, in $\mathcal{L}_d$ space, where each item are orthogonal with each other. Therefore, by Orthogonal Decomposition Theorem, $\mathcal{L}_d$ can be decomposed into the space spanned by $\mathbf{B}_n$ and the space orthogonal to this spanned space (or the space spanned by the rest frequencies component).

By using the Fourier series to expand the $y$, this can be decomposed into $y = y_{\mathbf{B}}^{\dagger} + y_{\mathbf{B}}$ by orthogonal decomposition theorem mentioned before.

$$\phi(\mathbf{y}, f(\mathbf{x}; \mathbf{W})) = ||\mathbf{y} - f(\mathbf{x}; \mathbf{W})||_2 \tag{46}$$

$$= ||\mathbf{y}_{\mathbf{B}}^{\dagger} + \mathbf{y_B} - f(\mathbf{x}; \mathbf{W})||_2 \tag{47}$$

$$\textbf{By using triangular inequality:} ||x + y|| - ||x|| \leq ||\mathbf{y}|| \tag{48}$$

$$\geq ||\mathbf{y}_{\mathbf{B}}^{\dagger}||_2 - ||f(\mathbf{x}; \mathbf{W}) - \mathbf{y_B}||_2 \tag{49}$$

By Theorem 4.1 in (Arora et al., 2019a), with probability $1 - \delta$, $\phi(\mathbf{y}, f(\mathbf{x}; \mathbf{W})) = \sqrt{\sum (1 - \eta\lambda_i)^{2k}\langle \mathbf{v}_i, \mathbf{y}\rangle^2} \pm \epsilon$. We can only decompose the latter part and obtain the proposed result. $\square$

**Lemma 5.** *For two frequencies $\alpha \in \mathbb{R}^d$ and $\beta \in \mathbb{N}^d$, the dot product between the trigonometric functions of two frequencies $cos(2\pi\alpha\mathbf{x})$, $sin(2\pi\alpha\mathbf{x})$ and $sin(2\pi\beta\mathbf{x})$, $cos(2\pi\beta\mathbf{x})$ can be written as either:*

$$(-1)^{\lceil\frac{d}{2}\rceil}\frac{1}{2}(\frac{1}{(2\pi)^d\prod_{i=1}^d[(\beta+\alpha_z)+\alpha_r]_i}(sin(2\pi\sum_i[(\beta+\alpha_z)+\alpha_r]_i) \tag{50}$$

$$-\sum_j sin(2\pi\sum_{i\in[d]/j}[(\beta+\alpha_z)+\alpha_r]_i) + \cdots + (-1)^d\sum_j sin(2\pi[(\beta+\alpha_z)+\alpha_r]_j) \tag{51}$$

$$\pm\frac{1}{(2\pi)^d\prod_{i=1}^d[(\beta-\alpha_z)-\alpha_r]_i}(sin(2\pi\sum_i[(\beta-\alpha_z)-\alpha_r]_i) \tag{52}$$

$$-\sum_j sin(2\pi\sum_{i\in[d]/j}[(\beta-\alpha_z)-\alpha_r]_i) + \cdots + (-1)^d\sum_j sin(2\pi[(\beta-\alpha_z)-\alpha_r]_j)) \tag{53}$$

*Or*

$$= (-1)^{\lceil\frac{d}{2}\rceil}\frac{1}{2}(\frac{1}{(2\pi)^d\prod_{i=1}^d[(\beta+\alpha_z)+\alpha_r]_i}(cos(2\pi\sum_i[(\beta+\alpha_z)+\alpha_r]_i) \tag{54}$$

$$-\sum_j cos(2\pi\sum_{i\in[d]/j}[(\beta+\alpha_z)+\alpha_r]_i) + \cdots + (-1)^d\sum_j cos(2\pi[(\beta+\alpha_z)+\alpha_r]_j) \tag{55}$$

$$\pm\frac{1}{(2\pi)^d\prod_{i=1}^d[(\beta-\alpha_z)-\alpha_r]_i}(cos(2\pi\sum_i[(\beta-\alpha_z)-\alpha_r]_i) \tag{56}$$

$$-\sum_j cos(2\pi\sum_{i\in[d]/j}[(\beta-\alpha_z)-\alpha_r]_i) + \cdots + (-1)^d\sum_j cos(2\pi[(\beta-\alpha_z)-\alpha_r]_j)) \tag{57}$$

*Depending on the odd or even of d and the combination of cosine and sine functions.*

*Proof.* Since $\alpha$ is in $\mathbb{R}^d$, we can decompose it into $\alpha = \alpha_z + \alpha_r$ where $\alpha_z \in \mathbb{Z}^d$ and $\alpha_r \in [0,1)^d$

$$\int_0^1 \cdots \int_0^1 cos(2\pi\alpha\mathbf{x})sin(2\pi\beta\mathbf{x})d\mathbf{x} = \int_0^1 \cdots \int_0^1 cos(2\pi(\alpha_z\mathbf{x} + \alpha_r\mathbf{x}))sin(2\pi\beta\mathbf{x})d\mathbf{x} \quad (58)$$

$$= \frac{1}{2}\int_0^1 \cdots \int_0^1 sin(2\pi(\beta\mathbf{x} + \alpha_z\mathbf{x} + \alpha_r\mathbf{x})) + sin(2\pi(\beta\mathbf{x} - \alpha_z\mathbf{x} - \alpha_r\mathbf{x}))d\mathbf{x} \quad (59)$$

$$= \frac{1}{2}\int_0^1 \cdots \int_0^1 sin(2\pi((\beta + \alpha_z)\mathbf{x} + \alpha_r\mathbf{x})) + sin(2\pi((\beta - \alpha_z)\mathbf{x} - \alpha_r\mathbf{x}))d\mathbf{x} \quad (60)$$

**if d is odd** $\quad (61)$

$$= (-1)^{\lceil\frac{d}{2}\rceil}\frac{1}{2}(\frac{1}{(2\pi)^d\prod_{i=1}^d[(\beta+\alpha_z)+\alpha_r]_i}(cos(2\pi\sum_i[(\beta+\alpha_z)+\alpha_r]_i) \quad (62)$$

$$- \sum_j cos(2\pi\sum_{i\in[d]/j}[(\beta+\alpha_z)+\alpha_r]_i) + \cdots + (-1)^d\sum_j cos(2\pi[(\beta+\alpha_z)+\alpha_r]_j) \quad (63)$$

$$+ \frac{1}{(2\pi)^d\prod_{i=1}^d[(\beta-\alpha_z)-\alpha_r]_i}(cos(2\pi\sum_i[(\beta-\alpha_z)-\alpha_r]_i) \quad (64)$$

$$- \sum_j cos(2\pi\sum_{i\in[d]/j}[(\beta-\alpha_z)-\alpha_r]_i) + \cdots + (-1)^d\sum_j cos(2\pi[(\beta-\alpha_z)-\alpha_r]_j)) \quad (65)$$

**if d is even** $\quad (66)$

$$= (-1)^{\lceil\frac{d}{2}\rceil}\frac{1}{2}(\frac{1}{(2\pi)^d\prod_{i=1}^d[(\beta+\alpha_z)+\alpha_r]_i}(sin(2\pi\sum_i[(\beta+\alpha_z)+\alpha_r]_i) \quad (67)$$

$$- \sum_j sin(2\pi\sum_{i\in[d]/j}[(\beta+\alpha_z)+\alpha_r]_i) + \cdots + (-1)^d\sum_j sin(2\pi[(\beta+\alpha_z)+\alpha_r]_j) \quad (68)$$

$$+ \frac{1}{(2\pi)^d\prod_{i=1}^d[(\beta-\alpha_z)-\alpha_r]_i}(sin(2\pi\sum_i[(\beta-\alpha_z)-\alpha_r]_i) \quad (69)$$

$$- \sum_j sin(2\pi\sum_{i\in[d]/j}[(\beta-\alpha_z)-\alpha_r]_i) + \cdots + (-1)^d\sum_j sin(2\pi[(\beta-\alpha_z)-\alpha_r]_j)) \quad (70)$$

And similar for other cases. $\qquad \square$

**Theorem 3.** *For a d-dimensional target function* $\mathbf{y}(\mathbf{x}) = \sum_{\mathbf{n}\in\mathbb{Z}^d}\hat{\mathbf{y}}_\mathbf{n}e^{i\mathbf{n}^\top\mathbf{x}}$, *where* $\hat{\mathbf{y}}_\mathbf{n}$ *are the corresponding coefficients of the Fourier series expansion of the* $\mathbf{y}(\mathbf{x})$. *Given a pre-sampled frequency set* $\mathbf{B}_n = \{\mathbf{b}_i \in \mathbb{R}^d\}_{i\in[N]}$ *and the* $L_2$ *loss function as* $\phi(\mathbf{y}, f(\mathbf{x};\mathbf{W})) = ||f(\mathbf{x};\mathbf{W}) - \mathbf{y}||_2$. *Then, for the frequency component* $\mathbf{n} \in \mathbb{Z}^d$, *and a sampled frequency* $\mathbf{b} \in \mathbf{B}_n$ *and its decomposition into the integer,* $\mathbf{b}_z$, *and residual part,* $\mathbf{b}_r \in [0,1)$, *the decreasing rate of the loss function for specific frequency* $\mathbf{n}$ *from the target function using two-layers MLPs with second layer frozen is* $\mathcal{O}(\frac{1}{\prod_{i=1}^d[(\mathbf{n}+\mathbf{b}_z)+\mathbf{b}_r]_i} + \frac{1}{\prod_{i=1}^d[(\mathbf{n}-\mathbf{b}_z)-\mathbf{b}_r]_i})$.

*Proof.* Again by Theorem 4.1 from (Arora et al., 2019a), we know that with probability $1-\delta$ $\sqrt{\sum(1-\eta\lambda_i)^{2k}\langle\mathbf{v}_i, y\rangle^2} \pm \epsilon = \phi(y, f(\mathbf{x};\mathbf{W}))$. By Lemma 5, we know that the inner product between the eigenfunctions of NTK and each integer frequency of the component of the decomposition of y is proportional to

$$\frac{1}{\prod_{i=1}^d[(\mathbf{n}+\mathbf{b}_z)+\mathbf{b}_r]_i}\varsigma(\mathbf{b}_r) + \frac{1}{\prod_{i=1}^d[(\mathbf{n}-\mathbf{b}_z)-\mathbf{b}_r]_i}\varsigma'(\mathbf{b}_r)$$

where $\varsigma(\mathbf{b}_r)$ is the sine/cosine function introduced in Lemma 5, and the integer terms, $\mathbf{b}_z$ and $\mathbf{n}$, can be ignored as they only contribute $2\pi n$ to the sine and cosine functions, which does not affect values of these periodic functions.

Since $\mathbf{b}_r$ is in [0,1) and is independent of the integer frequencies and we would like investigate how much will these non-integer frequencies activate each integer frequencies that consist in the Fourier series expansion of the target function. Therefore, we can consider the $\varsigma'(\mathbf{b}_r)$ and $\varsigma(\mathbf{b}_r)$ as

a constant term and only investigate the coefficient terms, which implies the result that decreasing rate is $\mathcal{O}(\frac{1}{\prod_{i=1}^{d}[(\mathbf{n}+\mathbf{b}_z)+\mathbf{b}_r]_i} + \frac{1}{\prod_{i=1}^{d}[(\mathbf{n}-\mathbf{b}_z)-\mathbf{b}_r]_i})$.

W.l.o.g. we can assume that $\mathbf{n} + \mathbf{b}_z \geq 0$, therefore, $\frac{1}{\prod_{i=1}^{d}[(\mathbf{n}+\mathbf{b}_z)+\mathbf{b}_r]_i} \geq 1$ iff $\mathbf{n} = \mathbf{b}_z = 0$ and $\frac{1}{\prod_{i=1}^{d}[(\mathbf{n}-\mathbf{b}_z)-\mathbf{b}_r]_i} \geq 1$ iff $\mathbf{n} = \mathbf{b}_z$. And since $\beta$ and $\alpha_z$ are integers, the difference is also integer. Therefore, if the difference is not zero, then it is bigger than 1 and leads to a large coefficient.

This implies the closer between the two frequencies, the larger the inner product. Further, we can deduce that the closer between two frequencies, the larger the decrease rate of the loss. $\qquad\square$

**Lemma 1.** *Considering two different Fourier features, Positional Encoding, and Random Fourier Features as in Def. 1. For two sampled frequencies using two embedding, $\mathbf{b}_{pe}$ from Positional Encoding and $\mathbf{b}_{rff}$ from Random Fourier Features, assume $\mathbf{b}_{rff}$ has two components with $[\mathbf{b}_{rff}]_1 \gg [\mathbf{b}_{rff}]_2$, and $\mathbf{b}_{pe}$ has only one non-zero component, $[\mathbf{b}_{pe}]_2$, equal to $[\mathbf{b}_{rff}]_2$. Let $\mathbf{b}_z$ be the closest integer frequency to $\mathbf{b}_{rff}$ and $[\mathbf{b}_{rff}]_2 = [\mathbf{b}_z]_2$. Then the decay rate of $\mathbf{b}_z$ for Positional Encoding, $\mathbf{b}_{pe}$, is equal to $[\mathbf{b}_z]_2$ for Random Fourier Features.*

*Proof.* By Theorem 3, we know that the decreasing rate for any integer frequency component $\mathbf{b}_z$ is proportional to $\mathcal{O}(\frac{1}{\prod_{i=1}^{d}[(\mathbf{n}+\mathbf{b}_z)+\mathbf{b}_r]_i} + \frac{1}{\prod_{i=1}^{d}[(\mathbf{n}-\mathbf{b}_z)-\mathbf{b}_r]_i})$. Denoting the $[\mathbf{v}]_i$ as the $i^{th}$ component of the vector $\mathbf{v}$.

For positional encoding, $\mathbf{b}_{pe}$ to learn the integer frequency $\mathbf{b}_z$, the decreasing rate is $\mathcal{O}(\frac{1}{[\mathbf{b}_z]_1+[\mathbf{b}_{pe}]_1+[\mathbf{b}_z]_2+[\mathbf{b}_{pe}]_2} + \frac{1}{[\mathbf{b}_z]_1-[\mathbf{b}_{pe}]_1+[\mathbf{b}_z]_2-[\mathbf{b}_{pe}]_2})$. Since the $[\mathbf{b}_{pe}]_2$ is zero and $[\mathbf{b}_{pe}]_1 = [\mathbf{b}_{rff}]_2$. Then the decreasing rate can be considered as $\mathcal{O}(\frac{1}{[\mathbf{b}_z]_1+[\mathbf{b}_{rff}]_2+[\mathbf{b}_z^2]_2} + \frac{1}{[\mathbf{b}_z]_1-[\mathbf{b}_{rff}]_2+[\mathbf{b}_z]_2})$

For Random Fourier Features, $\mathbf{b}_{rff}$ to learn $[\mathbf{b}_z]_2$, the decreasing rate is also $\mathcal{O}(\frac{1}{[\mathbf{b}_{rff}]_1+[\mathbf{b}_z]_2+[\mathbf{b}_{rff}]_2} + \frac{1}{-[\mathbf{b}_{rff}]_1+[\mathbf{b}_z]_2-[\mathbf{b}_{rff}]_2})$. Since $\mathbf{b}_{rff}]_1$ equals to $\mathbf{b}_z]_1$, this proofs the lemma.

$\qquad\square$

## A.4 LINE-SEARCH METHOD

Considering a minimization problem as the following:

$$\theta^* = \min_{\theta \in \Theta} f(\mathbf{X}; \theta) \tag{71}$$

In the context of machine learning $f(\cdot)$ usually denotes the loss function, $\theta$ represents the parameters of the machine learning algorithms that belong to parameter space $\Theta$ and $\mathbf{X}$ denotes the training dataset. One common method to find the $\theta^* \in \Theta$ that minimizes $f(\cdot)$ is to use the gradient descent method as shown in the Algorithm 1.

---
**Algorithm 1** Gradient Descent Algorithm

1: **Initialize:** variables $\theta_0$, max iteration $N$, learning rate $\alpha$
2: **for** $i \leftarrow 1$ to $N$ **do**
3:      Calculate the derivative of $f(\theta_{t-1})$ about $\theta_{t-1}$ as direction denotes as $p_t$
4:      $\theta_t \leftarrow \theta_{t-1} + \alpha p_t$
5: **end for**

---

Based on this Gradient Descent method, the line-search method is to find the proper learning rate $\alpha_t$ at each iteration by optimization to solve the approximate learning rate or exact learning rate if possible. The algorithm can be shown as the Algorithm 2.

## A.5 DETAILS OF CUSTORM LINE-SEARCH ALGORITHM

In this section, we will explain the derivation of the modified line-search algorithm used to determine the learning rate of the adaptive filter.

---

**Algorithm 2** Line-search Method

---

1: **Initialize:** variables $\theta_0$, max iteration $N$
2: **for** $i \leftarrow 1$ to $N$ **do**
3:    Calculate the derivative of $f(\theta_{t-1})$ about $\theta_{t-1}$ as direction denotes as $p_t$
4:    $\alpha = arg\,min_{\alpha_t} f(\theta_{t-1} + \alpha p_t)$
5:    $\theta \leftarrow \theta_{t-1} + \alpha_t p_t$
6: **end for**

---

Let $f(\theta_A^t, \theta_I^t)$ denote the loss function at iteration $t$, $\mathbf{p}_A^t$ as the update direction for the adaptive filter, and $\mathbf{p}_I^t$ as the update direction for the INRs. We can then perform a Taylor expansion around the parameters $(\theta_A^{t-1}, \theta_I^{t-1})$, expressed as follows:

$$f(\theta_A^t, \theta_I^t) = f(\theta_A^{t-1}, \theta_I^{t-1}) - \nabla_{\theta_A^t} f(\theta_A^{t-1}, \theta_I^{t-1})^\top (\theta_A^t - \theta_A^{t-1}) - \nabla_{\theta_I^t} f(\theta_A^{t-1}, \theta_I^{t-1})^\top (\theta_I^t - \theta_I^{t-1})$$

$$+\frac{1}{2}\left[(\theta_A^t - \theta_A^{t-1})^2 \triangle_{\theta_A^t} f(\theta_A^{t-1}, \theta_I^{t-1}) + (\theta_I^t - \theta_I^{t-1})^2 \triangle_{\theta_I^t} f(\theta_A^{t-1}, \theta_I^{t-1})\right] + \mathcal{O}((\theta_I^t - \theta_I^{t-1})^2, (\theta_A^t - \theta_A^{t-1})^2)$$

Using the gradient descent method, we have $\theta^t = \theta^{t-1} + \alpha p^{t-1}$. Since the ReLU activation function results in the second and higher-order derivatives being zero, the equation simplifies to:

$$f(\theta_A^t, \theta_I^t) \approx f(\theta_A^{t-1}, \theta_I^{t-1}) - \nabla_{\theta_A^t} f(\theta_A^{t-1}, \theta_I^{t-1})^\top (\alpha_A \mathbf{p}_A^{t-1}) - \nabla_{\theta_I^t} f(\theta_A^{t-1}, \theta_I^{t-1})^\top (\alpha_I \mathbf{p}_I^{t-1})$$

$$\phi(\alpha_A) = f(\theta_A^t, \theta_I^t) \approx k\alpha_A + b$$

$$where \, k = -\nabla_{\theta_A^t} f(\theta_A^{t-1}, \theta_I^{t-1})^\top \mathbf{p}_A^{t-1} \textbf{ and } b = f(\theta_A^{t-1}, \theta_I^{t-1}) - \nabla_{\theta_I^t} f(\theta_A^{t-1}, \theta_I^{t-1})^\top (\alpha_I \mathbf{p}_I^{t-1})$$

Since the learning rate of the INRs part is known, this can be simplified as a linear optimization problem with only an order 1 unknown parameter $\alpha_A^t$.

$$\arg\min_{\alpha_A} f(\theta_A^t, \theta_I^t) \approx \arg\min_{\alpha_A} f(\theta_A^{t-1}, \theta_I^{t-1}) - \nabla_{\theta_A^t} f(\theta_A^{t-1}, \theta_I^{t-1})^\top (\alpha_A \mathbf{p}_A^{t-1}) - \nabla_{\theta_I^t} f(\theta_A^{t-1}, \theta_I^{t-1})^\top (\alpha_I \mathbf{p}_I^{t-1})$$

Furthermore, to prevent the impact of a small denominator, we add a constant $\epsilon = 1 \times 10^{-6}$ to ensure the robustness of the algorithm. The solution to this optimization problem can be determined through case analysis by examining the sign of the slope and intercept, as illustrated in Figure 7.

The overall algorithm pipeline is shown in the following Algorithm 3

To ensure sufficient decrease, we also apply a similar derivation based on the Armijo condition, which is commonly used in line-search algorithms to guarantee sufficient decrease. The Armijo condition is typically expressed as follows:

$$f(x_k + \alpha_k p_k) \leq f(x_k) + c_1 \alpha_k p_k^\top \nabla f(x_k),$$

Where $c_1$ typically takes the value $1 \times 10^{-3}$. By assuming the current step is $t$ (which is equal to the previous derivation's $t-1$, but we denote it as $t$ for simplicity), this can be written as:

$$f(\theta_A^t + \alpha_A \mathbf{p}_A^t, \theta_I^t + \alpha_I \mathbf{p}_I^t) \leq f(\theta_A^t, \theta_I^t) + c_1 \alpha_A p_k^\top \nabla_{\theta_A^{t-1}} f^\top \mathbf{p}_A^t + c_1 \alpha_I \nabla_{\theta_I^{t-1}} f^\top \mathbf{p}_I^t$$

Therefore, using a similar Taylor expansion on the loss function with respect to the parameters and algorithm is shown in Algorithm.4:

$$f(\theta_A^t, \theta_I^t) - \nabla_{\theta_A^t} f(\theta_A^t, \theta_I^t)^\top (\alpha_A \mathbf{p}_A^t) - \nabla_{\theta_I^t} f(\theta_A^t, \theta_I^t)^\top (\alpha_I \mathbf{p}_I^t) \leq f(\theta_A^t, \theta_I^t) + c_1 \alpha_A p_k^\top \nabla_{\theta_A^t} f^\top \mathbf{p}_A^t + c_1 \alpha_I \nabla_{\theta_I^t} f^\top \mathbf{p}_I^t$$

$$(c_1 - 1)\alpha_I \nabla_{\theta_A^t} f^\top \mathbf{p}_A^t \geq (1 - c_1)\nabla_{\theta_I^t} f^\top \mathbf{p}_I^t$$

---

**Algorithm 3** Line-search Method-Relative Learning Rate

---

1: **Initialize:** variables $\theta_0$, max iteration $N$, $\alpha_A$, $\alpha_I$, $\alpha_{max}$, $\alpha_{min}$, $\epsilon$, $c_1$
2: **for** $i \leftarrow 1$ to $N$ **do**
3:     Calculate the update direction as $\mathbf{p}_A^t$ and $\mathbf{p}_I^t$
4:     Calculate the partial derivative of $f(\theta_A^{t-1})$ about $\theta_A^{t-1}$ as direction denotes as $\nabla_{\theta_A^{t-1}} f$
5:     Calculate the partial derivative of $f(\theta_I^{t-1})$ about $\theta_I^{t-1}$ as direction denotes as $\nabla_{\theta_I^{t-1}} f$
6:     $k \leftarrow -\nabla_{\theta_A^{t-1}} f(\theta_A^{t-1}, \theta_I^{t-1})^\top \mathbf{p}_A^{t-1} + \epsilon$
7:     $b \leftarrow f(\theta_A^{t-1}, \theta_I^{t-1}) - \alpha_I \nabla_{\theta_I^{t-1}} f^\top \mathbf{p}_I^t$
8:     **if** $a \geq 0, b \leq 0$ **then**
9:         $\alpha_A \leftarrow \alpha_{min}$
10:         **Armijo Condition Check**$(c_1, \nabla_{\theta_A^{t-1}} f^\top \mathbf{p}_A^t, \nabla_{\theta_I^{t-1}} f^\top \mathbf{p}_I^t)$
11:     **else if** $a \geq 0, b \geq 0$ OR $a \leq 0, b \leq 0$ **then**
12:         $\alpha_A \leftarrow Clip \left[ |\frac{-b}{k}|, \alpha_{min}, \alpha_{max} \right]$
13:         **Armijo Condition Check**$(c_1, \nabla_{\theta_A^{t-1}} f^\top \mathbf{p}_A^t, \nabla_{\theta_I^{t-1}} f^\top \mathbf{p}_I^t)$
14:     **else**
15:         $\alpha_A \leftarrow \alpha_{min}$
16:         **Armijo Condition Check**$(c_1, \nabla_{\theta_A^{t-1}} f^\top \mathbf{p}_A^t, \nabla_{\theta_I^{t-1}} f^\top \mathbf{p}_I^t)$
17:     **end if**
18:     $\theta_A^t \leftarrow \theta_A^{t-1} + \alpha_A \mathbf{p}_A^t$
19:     $\theta_I^t \leftarrow \theta_I^{t-1} + \alpha_I \mathbf{p}_I^t$
20: **end for**

---

**Algorithm 4** Armijo Condition Check

---

**Initialize:** variables $c_1, \nabla_{\theta_A^{t-1}} f^\top \mathbf{p}_A^t, \nabla_{\theta_I^t} f^\top \mathbf{p}_I^t$, current step t
**if** $(c_1 - 1)\nabla_{\theta_A^t} f^\top \mathbf{p}_A^t > (1 - c_1)\nabla_{\theta_I^t} f^\top \mathbf{p}_I^t$ **then**
    **if** $\nabla_{\theta_I^t} f^\top \mathbf{p}_I^t \times \nabla_{\theta_A^{t-1}} f^\top \mathbf{p}_A^t > 0$ **then**
        **return** $\alpha_A \leftarrow \dfrac{\nabla_{\theta_I^t} f^\top \mathbf{p}_I^t}{\nabla_{\theta_A^t} f^\top \mathbf{p}_A^t}$
    **else**
        **return** $\alpha_A \leftarrow \alpha_{min}$
    **end if**
**else**
    **return** None
**end if**

---

## A.6 VISUALIZATION OF THE FINAL OUTPUT OF FILTERS

In this section, we further present the results of the filtered Positional Encoding embedding. Compared to Random Fourier Features, which involve more complex combinations of frequency components, Positional Encoding displays more regular frequency patterns, making it better suited for visualization. These visualizations demonstrate that in low-frequency regions, the high-frequency embeddings are effectively suppressed by the filter, in line with our expectations of the adaptive linear filter's behavior. Additionally, for low-frequency embeddings, the filter can also emphasize high-frequency components, enabling more fine-grained outputs.

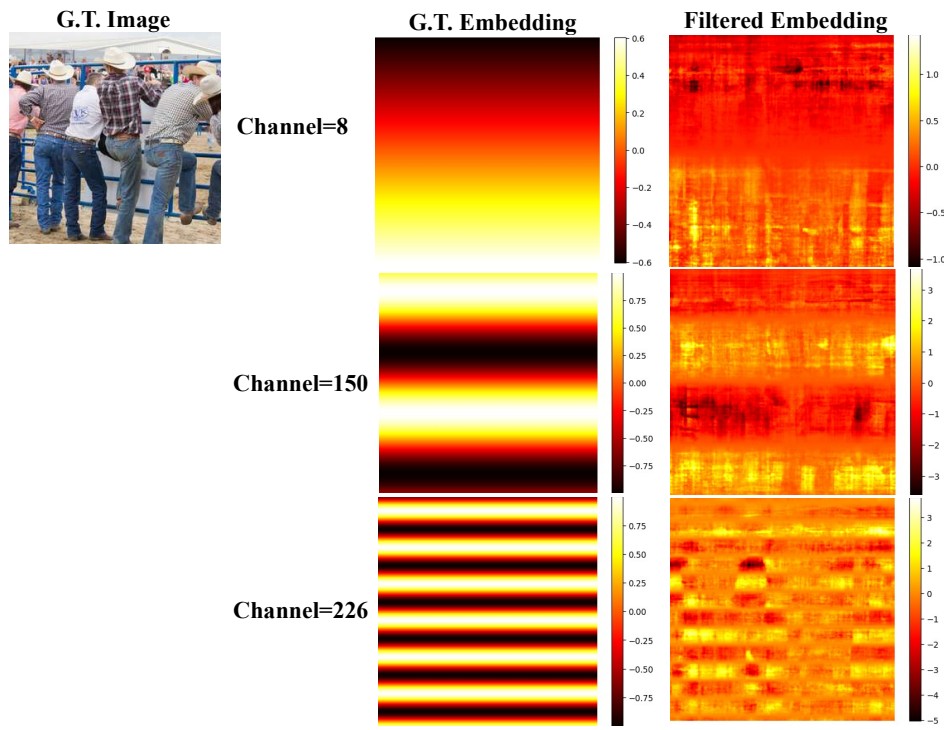

Figure 13: Visualization of the filtered embedding for image 804 in the DIV2K validation split.

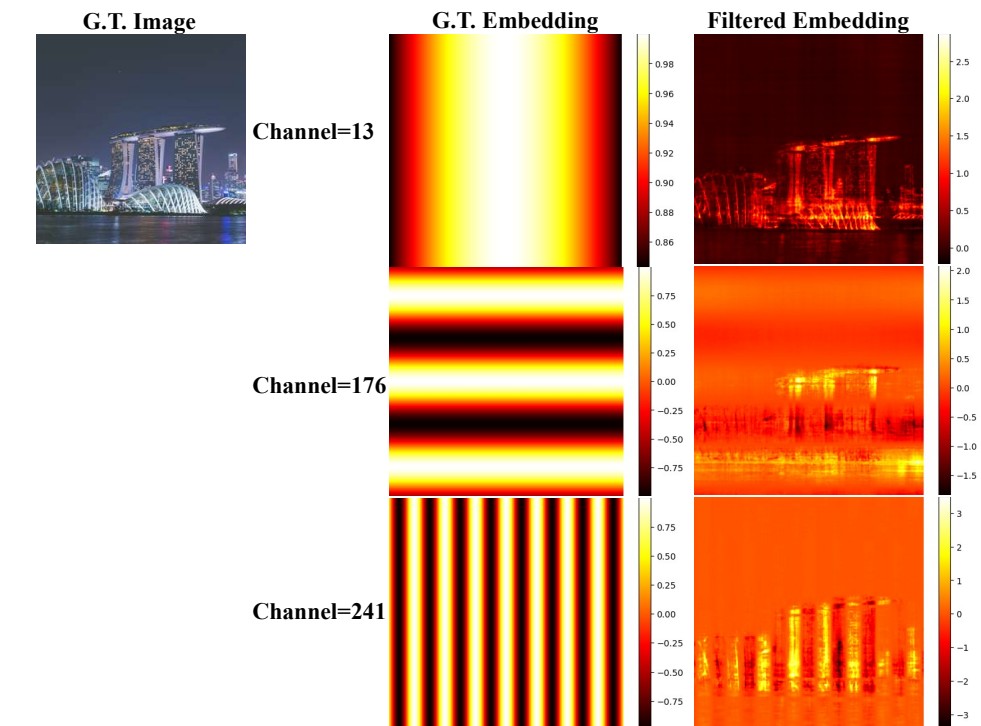

Figure 14: Visualization of the filtered embedding for image 814 in the DIV2K validation split.

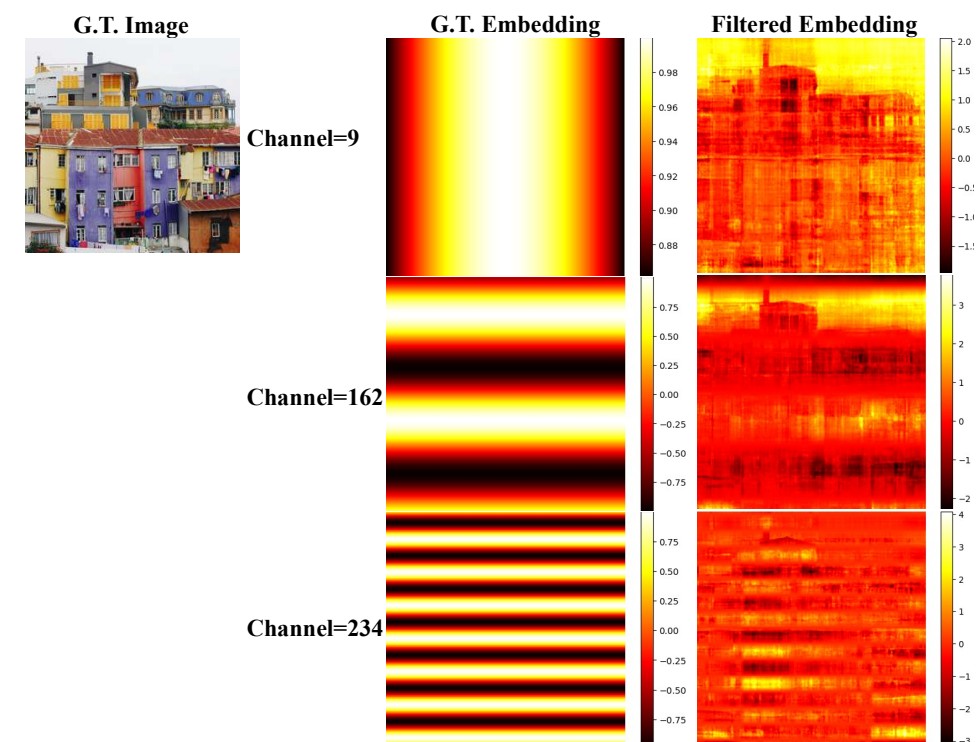

Figure 15: Visualization of the filtered embedding for image 818 in the DIV2K validation split.

## A.7 FURTHER EXPERIMENT VISUALIZATION

### A.7.1 3D SHAPE REGRESSION

In this section, we further provide some visualizations of the fitted result of 3D shape regression to provide a more detailed idea of the performance of our proposed method.

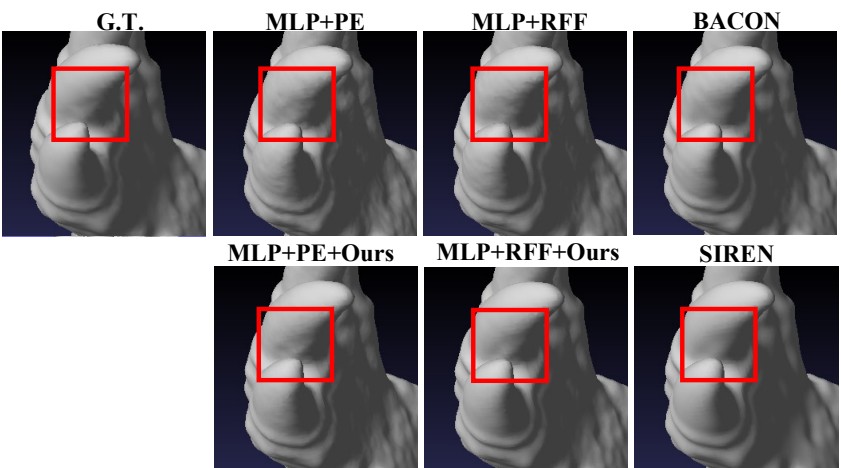

Figure 16: Visualization of the Mesh of Armadillo in the Stanford 3D Scanning Repository.

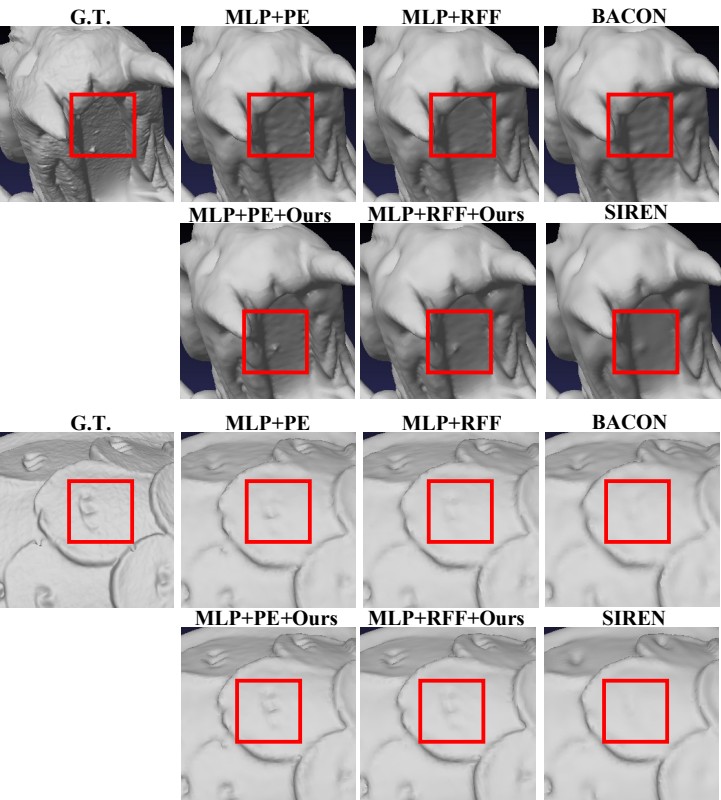

Figure 17: Visualization of the Mesh of Dragon in the Stanford 3D Scanning Repository.

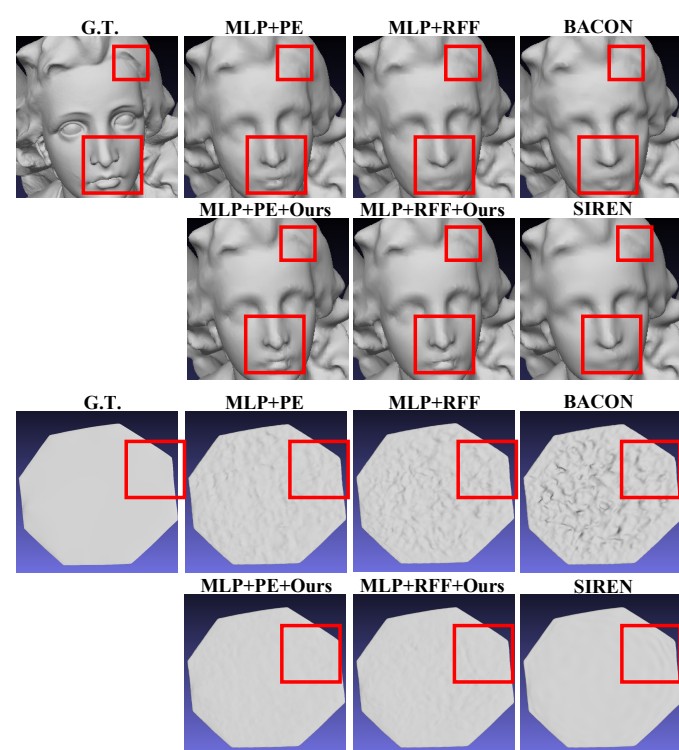

Figure 18: Visualization of the Mesh of Lucy in the Stanford 3D Scanning Repository.

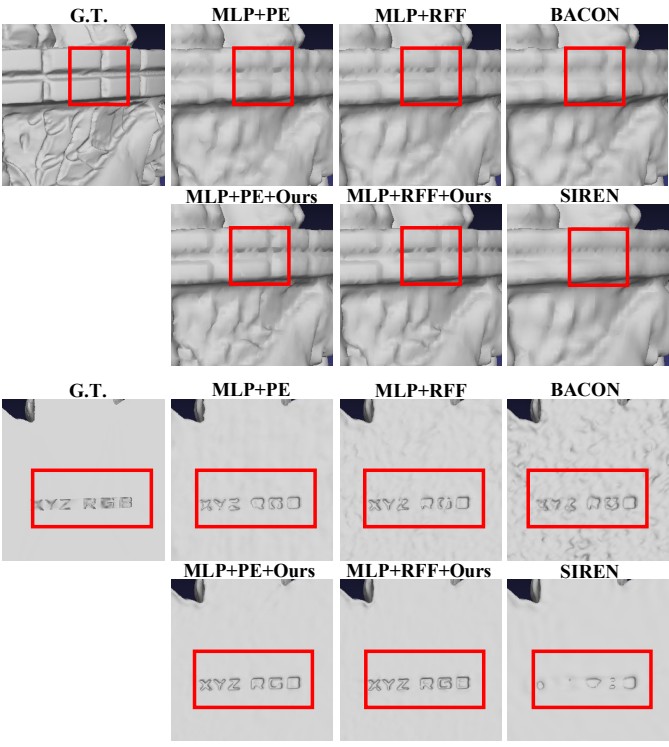

Figure 19: Visualization of the Mesh of Thai in the Stanford 3D Scanning Repository.

## A.7.2 NEURAL RADIANCE FIELD

Figure 20: Further Visualization of the NeRF task.

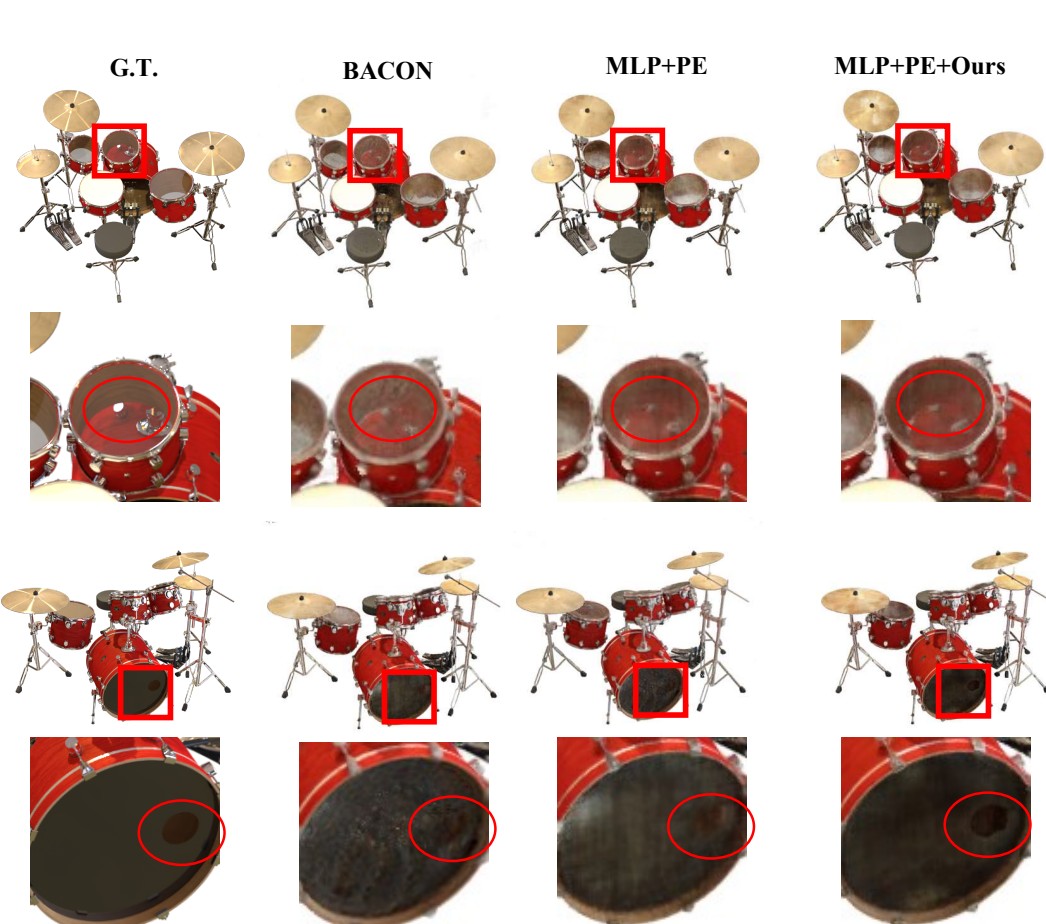

Figure 21: Further Visualization of the NeRF task.

