# OpenReview forum: "Neural Tangent Kernel Analysis and Filtering for Robust Fourier Feature Embedding"
_ICLR.cc/2025/Conference — ICLR 2025 Conference Withdrawn Submission_

### Official Review · Reviewer_Ksgq · 2024-10-26

**Soundness:** 2
**Presentation:** 2
**Contribution:** 3
**Rating:** 3
**Confidence:** 4

**Summary:**

The paper points out a limitation in Fourier feature embeddings popular in implicit neural representations (INRs), and proposes a strategy to partially address this limitation. The limitation is essentially that a finite Fourier series will not perfectly capture a constant region of a signal, but will instead have some high-frequency noise in these low-frequency regions (where high-frequency means the highest frequencies in the model). The proposed mitigation strategy is to introduce an MLP without bias as a piecewise linear filter, allowing the INR to use different frequency components to represent different regions of a signal. This adaptive filter can then suppress high-frequency noise in low-frequency regions without causing blurring in high-frequency regions. The paper provides a mixture of NTK analysis and experiments on 2D image fitting, 3D radiance field modeling, and 3D signed distance field modeling.

**Strengths:**

- The paper points out a valid limitation of existing INRs based on Fourier features, and the proposed method shows improvement over the current methods based on positional encoding and random Fourier features.
- It’s not stated as clearly as I would like, but I think the proposed approach gets at an interesting point, namely that the representation power of Fourier series is complementary to the representation power of relu networks, so it makes sense to expect improvement by combining them. For example, a rectangular function (value 1 within an interval and value 0 outside) is easily represented by a small relu MLP, but would require an infinite Fourier series to represent exactly. Conversely, a sum of sinusoids is easy to represent in the Fourier basis with a few components, but would require infinite piecewise-linear regions to approximate with a relu MLP. I think a revised paper that explores this idea more clearly and thoroughly could be very impactful.
- I appreciate that the paper includes experiments on 3 separate tasks, showing generality of their proposed INR. In particular, the results on the NeRF task are most compelling.

**Weaknesses:**

- The exposition of the paper tends to obscure the main ideas rather than elevating and clarifying them. Please refer to the separate list of presentation concerns below; although each concern alone is minor, together they severely lessen the potential impact of the paper.
- The experiments for image fitting are in the overparameterized regime. The number of unknowns is 256*256*3, and the number of parameters just in the adaptive linear filter (not even counting the rest of the INR layers) is at least 384*384*3. All of the INRs in this comparison would be outperformed by a pixel grid, which could achieve zero error with fewer parameters. I would find the experiment much more convincing if the models were actually doing compression, and if the number of parameters were held constant across all the models being compared.
- The results in Table 2 are very similar across all models, as are the visual results in Figure 9; for shape regression I don’t see a clear improvement from using the proposed adaptive linear filter, especially since it likely also increases model size and training time.
- The ablation study in Table 4 looks like the vast majority of the benefit from the proposed line search is in image regression, which as I mentioned above is not a compelling setting due to the model overparameterization. I would suggest checking if the line search is still valuable for larger images (underparameterized/compression regime), and if not then the model might be simpler and essentially as good without it.

presentation concerns & suggestions
- The abstract and introduction make a big deal of the proposed adaptive linear filter being “bias-free”, but the explanations of what type of bias this is referring to (it turns out to be the additive term in the linear layer of an MLP, rather than e.g. some subgroup bias in the data) and why bias-free is desirable are deferred to page 6. Once we reach this explanation, it provides claims about why bias-free layers are desirable but doesn’t provide any evidence to justify these claims.
- The paper repeatedly refers to a Pareto efficiency tradeoff between the two components of the model (the INR and the new adaptive linear filter). The issue is that this is not what Pareto efficiency means…Pareto efficiency refers to a setting where there are two objective functions, not two components that need to be in balance to achieve a single objective. For example, there might be a Pareto efficiency tradeoff between memory and quality. However, in this paper only quality metrics are considered so there is no notion of a Pareto frontier.
- Frequency-based embeddings (random Fourier features and positional encoding) are repeatedly referred to as projections. Please use a term like “embedding” or “encoding” rather than projection, because these are not projections in the linear algebra sense of the term.
- There are many places where grammar is unclear, including some sentence fragments. See e.g. line 044-045, line 188, and the text in all of the theorem statements.
- Figures often have important information (e.g. tick marks, axis labels, color bars) in font that would be illegible without zooming in. Not all readers will be able or willing to zoom in (e.g. if they read a printed copy).
- Figure 1: the scales for the y axes should match, so that the subfigures for random Fourier features and positional encoding can be visually compared. Also for this figure, the description in the caption is not obviously in agreement with the data shown in the figure (i.e. the caption claims that specifically the model with positional encoding struggles to fit high frequency signal, but the maximum values for PSNR of high frequency signal—the red curves—are very similar for both models).
- The related work section lists and lightly summarizes related work, but does not explain the strengths and weaknesses of prior strategies or how they relate to the current proposed method. Each subsection should end with some sort of concluding sentence or two with this information.
- The theorems are not clear. The notation is not always defined (for example, there are multiple variables k in theorem 1, and I suspect they might refer to different things). Also in theorem 1 the first line of math could be simplified with a cosine difference identity, as was done in the paper introducing random Fourier features…is there a reason to leave it in the unsimplified form?
- Every theorem statement should be (1) self-contained, so that all notation is defined inside the theorem statement, and (2) preceded or followed by some explanation or interpretation of the results. A little interpretation is provided before the theorems, but more is needed especially to distinguish each theorem from the others. Perhaps Figure 4 could also be shown in conjunction with the theory, as an illustration, instead of putting it in the methods section.
- Figure 4 could be improved by showing that this well-known phenomenon from Fourier series approximation also occurs with INRs.
- Figure 7 is not very clear…I would suggest either adding more description of this line search method in the main paper rather than the appendix, or moving this figure to the appendix also, rather than separating the algorithm description from the figure.
- What is the color scale of the image in Figure 8? I see that the color bar for the error maps goes between 0 and 1, but I don’t know what the original dynamic range of the image is. If the original image is also in the range [0,1] then I’m quite surprised to see existing methods with error approaching 1.
- Table 3 appears before Table 2.
- The description of figure 10 on line 446-447 sounds like it is describing the Lego scene rather than the Microphone scene that is actually shown. That said, the improvement on the Mic is visually compelling.

**Questions:**

Please refer to questions embedded in the weaknesses section.

---

### Official Review · Reviewer_5f7e · 2024-10-27

**Soundness:** 3
**Presentation:** 3
**Contribution:** 3
**Rating:** 3
**Confidence:** 5

**Summary:**

An interesting Neural Tangent Kernel-based approach is proposed for INR for dealing with the problem of spectral bias.  In particular, the proposed method incorporates  a line-search algorithm to achieve a Pareto-efficient balance between frequency learning and noise reduction.  Various experiments are conducted to show the significance of this work.

**Strengths:**

+ Well-written paper.
+ From Neural Tangent Kernel (NTK), the authors analyze the high-frequency noise in Fourier Features and it is shown that it arises due to limited frequency sampling.
+An interesting line-search algorithm is proposed to achieve a Pareto-efficient balance between frequency learning and noise reduction.
+ Many experiments are conducted on various modalities to show the significance of this work.

**Weaknesses:**

- Since the overall approach is for improving INRs, comparison with the latest methods such as WIRE, and Gaussian-based INRs are missing.  Without proper comparisons with those latest methods, one can not see the significance of the proposed method.
-Another important experiment missing from the paper is on restoration.  How well does the proposed method work on various restoration tasks such as inpainting, denoising, super-resolution?

**Questions:**

Please see my detailed comments above.

---

### Official Review · Reviewer_aZ4z · 2024-11-02

**Soundness:** 2
**Presentation:** 2
**Contribution:** 2
**Rating:** 3
**Confidence:** 5

**Summary:**

This paper addresses the resultant noise introduced by Fourier features-based methods such as positional encoding. Using the Neural Tangent Kernel (NTK) analysis, the authors theoretically explain the generation of high-frequency noise when employing Fourier features embedding. To address this issue, they propose a bias-free MLP as an adaptive linear filter to suppress unnecessary high frequencies and a line-search-based optimization method to dynamically adjust the learning rate of the filter. The proposed method improves the performance of INRs across various tasks marginally, including image regression, 3D shape regression, and Neural Radiance Fields (NeRF), achieving better noise reduction and detail preservation compared to existing methods.

**Strengths:**

1.	The paper provides a theoretical analysis of the high-frequency noise introduced by Fourier features embedding (although I do not know what are the relationship connected with the following methods), and proposes a sample bias-free MLP with a line-search-based optimization to alleviate this problem.
2.	Marginal improvements are achieved in the classical INR’s tasks.

**Weaknesses:**

1.	The proposed method of masking high-frequency embeddings in low-frequency regions may lack sufficient innovation. The idea is similar to SAPE (Spatially-Adaptive Progressive Encoding for Neural Optimization), which applies an adaptive encoding layer to mask encoded features.
2.	The theoretical analysis emphasizes that finite sampling introduces noise primarily due to unsampled frequencies. However, in practice, the primary noise may be introduced by sampled high frequencies.
For example, a discrete signal or a simple function composed of sine and cosine functions can be decomposed by finite frequencies, however, noise may still appear in the neural network outputs, even if many frequencies are sampled.
Additionally, the proposed adaptive linear filter to alleviate high-frequency noise is based on the latter theory, not former, resulting in a mismatch between the theoretical analysis section and the methods section.
3.	Most of INR papers conduct experiment on image representation using the DIV2K dataset with a 1024$\times$1024 resolution, however this paper only provides results on images with a 256$\times$256 resolution. In the experiment, the authors report ~50dB PSNR values achieved by the proposed method, which are surprising and hinder a fair comparison with the PSNR values reported by other INRs, such as WIRE and FINER (both are not cited and compared in the paper).
4.	The paper ‘On the frequency-bias of coordinate-mlps’ focuses on a same problem, however it is not cited and compared.
5.	What impact do adaptive linear filters of different sizes (width, number of layers) have on training results and stability?
6.	Since the bias-free MLP introduces additional learnable parameters compares to baselines, what are the experimental results when baseline models are provided with comparable parameters? Furthermore, the term 'bias-free MLP' used in this paper can be misunderstanding, as it might give the impression that it refers to a MLP without spectral-bias (notably, the term ‘spectral bias’ is also mentioned in the paper, deepening the misunderstanding).
7.	Given that the proposed method balances noise reduction with the preservation of high-frequency details, could it potentially demonstrate superior performance in interpolation tasks, such as image generalization?

**Questions:**

see weakness.

---

### Official Review · Reviewer_Bhax · 2024-11-03

**Soundness:** 2
**Presentation:** 2
**Contribution:** 2
**Rating:** 5
**Confidence:** 3

**Summary:**

This paper propose that Fourier features-based INRs methods will introduce the noise due to the inadequacy of its limited sampling frequency. Based on this perspective, this paper proposes bias-free MLPs to locally adjust the frequency of Fourier embeddings at the coordinate level. This paper gives a theoretical analysis on the noise generation. This paper validates the bias-free MLPs on image regression, 3D shape regression and NeRF.

**Strengths:**

The idea of this paper that suppress the unnecessary frequencies and amplify the essential frequencies in Fourier features is novel. This paper addresses the frequency selection and overall model optimization issues by introducing bias-free MLPs and dynamic learning rates, achieving Pareto efficiency. This paper also contains a comprehensive theoretical analysis and well-rounded experiments.

**Weaknesses:**

1.	The connection between the theoretical analysis and the proposed method in the paper should be better refined and clarified to avoid the theoretical analysis appearing contrived.

The overall paper appears to have little connection with NTK theory, and excessive mention of NTK makes the presentation of the paper’s main theme unclear. Specifically, the insight of the adaptive linear filter mainly depends on the Theorem 2. However, the proof of the Theorem 2 does not seem to require NTK theory and the core proof of Theorem 1. Then, Theorem 1 also seems to be meaningless for this method. Please give more illustration about the insights of Theorem 1. There are the same problems in the Theorem 3 and Lemma 1. Based on these potential concerns, the related work of NTK should be also refined and add more recent works such training dynamics methods for spectral bias.

2.	The details of the bias-free MLPs for adaptive linear filter should be clarified. Concretely, although the paper （line 294-296）give a simplified description, a concise mathematical formulation are more clear and precise. Further, the visualization of the filtered results （line 306, Figure 6, 13-15）should be explained, as the dimensionality of the Fourier embedding is “H*W*embedding channel”.

3.	The experimental setup seems to be unfair or insufficiently trained, and the improvements are limited. For example, as the paper states that the experimental settings of 3D shape representation follow the BACON, the results of MLP+PE reported in Table 2 are only 0.96189 in IOU. However, the IOU in supplemental results of BACON is around 0.980 in IOU. Further, improvements of the adaptive linear filter are very minimal in 3D shape representations.

**Questions:**

Can this method be applied to larger signals such as the full-resolution of DIV2K images? If possible, what are its runtime and memory usage like?

---

### Note · Authors · 2024-11-13

I have read and agree with the venue's withdrawal policy on behalf of myself and my co-authors.